# Mixed trends in heavy metal-enriched fugitive dust on National Park Service lands along the Red Dog Mine haul road, Alaska, 2006–2017

Peter N. Neitlich[1]*, Wilson Wright[2], Elisa Di Meglio[3], Alyssa E. Shiel[4], Celia J. Hampton-Miller[5], Mevin B. Hooten[6]

1 Alaska Regional Office-Natural Resources Team, National Park Service, Anchorage, Alaska, United States of America, 2 Department of Statistics, Colorado State University, Fort Collins, Colorado, United States of America, 3 Department of Botany, Oregon State University, Corvallis, Oregon, United States of America, 4 College of Earth, Ocean, and Atmospheric Sciences, Oregon State University, Corvallis, Oregon, United States of America, 5 Arctic Network, National Park Service, Fairbanks, Alaska, United States of America, 6 Department of Statistics and Data Sciences, The University of Texas at Austin, Austin, Texas, United States of America

* peter_neitlich@nps.gov

**Data Availability Statement:** Data is available at: https://irma.nps.gov/DataStore/Reference/Profile/2296579.

## Abstract

This study presents the status and trends of long-term monitoring of the elemental concentrations of zinc (Zn), lead (Pb), and cadmium (Cd) in *Hylocomium splendens* moss tissue in Cape Krusenstern National Monument (CAKR), Alaska, adjacent to the Red Dog Mine haul road. Spatial patterns of the deposition of these metals were re-assessed for the period from 2006–2017 following an identical study that assessed trends between 2001–2006. In contrast to the widespread and steep declines in Zn and Pb levels throughout most of the study area between 2001–2006, this study showed more mixed results for 2006–2017. At distances within 100 m of the haul road, only Pb decreased between 2006–2017. At distances between 100–5,000 m, however, both Zn and Cd decreased between 2006–2017, with high probabilities of decrease and percent decreases of 11–20% and 46–52% respectively. Lead did not decrease in any of the more distant areas. Following earlier work on lichen species richness in the study area, it appears that 2017 Zn levels are approaching those associated with "background" lichen species richness throughout a relatively large proportion of the study area at least 2,000 m from the haul road and several km from the port site. The findings in this study may be used to plan additional mitigation measures to reduce Zn deposition related to impacts on lichen communities.

## Introduction

Cape Krusenstern National Monument (CAKR) was created in 1980 [1] as a US National Park Service (NPS) unit. Located above the Arctic Circle north of Kotzebue, Alaska (Fig 1) the monument features low windswept mountains along the Chukchi Sea coast, a series of lagoons that open seasonally to the sea and feature a huge variety of fish and bird habitats, extensive soft-sediment shorelines, and a wide coastal plain [2]. In 1985, Congress authorized an industrial

**Funding:** This research was funded by the National Park Service, Arctic Network (Inventory and Monitoring Program).The funders had no role in study design, data collection and analysis, decision to publish, or preparation of the manuscript.

**Competing interests:** The authors have declared that no competing interests exist.

easement through the monument allowing for the trucking of metal ore concentrates from the Red Dog Mine to the Red Dog Port [3]. Both the mine and the port are located on Northwest Alaska Native Association (NANA) lands, and are linked via the Delong Mountain Transportation System (DMTS) haul road (referred to hereafter as "haul road"). The haul road crosses 32 km of NPS lands. Since 1989, the mine has transported Zn and Pb concentrates (approximately 55% in fine powder form) year-round from the mine to large concentrate storage buildings at the port. In doing so, fugitive dusts enriched with Zn, Pb, and Cd have been dispersed onto NPS lands both inside and outside of the industrial easement [4–6].

The moss *Hylocomium splendens* (Hedw.) Schimp has been used as a biomonitor of heavy metals, atmospheric pollutants, and even anthropogenic substances for over 30 years and mosses in general have been used for this purpose since the 1960s [7–9]. Dozens of studies have utilized the low cost and high repeatability of moss biomonitoring as an alternative to instrumented monitoring (see [7] for review), and this approach has been effective for both temporal and spatial analysis of air pollution deposition [10, 11]. While comparisons of elemental concentrations in moss tissue across and between broad regions (e.g., northern Europe, North America) can be partly confounded by environmental and depositional particularities, moss biomonitoring is widely known for high accuracy and consistency on local to smaller regional scales [12].

There is a large volume of literature on biomonitoring of heavy metal pollution related to metal smelting (e.g., [13–17]), but studies on biomonitoring of fugitive dust from mining operations are far fewer. While smelters are often responsible for acute effects on vegetation in a localized area, fugitive dusts can cause chronic effects on a landscape level for hundreds of km$^2$ or more [18]. When moss biomonitoring for contaminant deposition is paired initially with monitoring of biological effects, simple moss biomonitoring may then be used on a regional level to predict biological consequences.

NPS first measured the elemental concentrations of Zn, Pb, and Cd in the moss *Hylocomium splendens* in 2000 [6] along six transects from the DMTS haul road. Sampling was expanded to a broader area of inference using a spatial model of elemental deposition patterns in 2001 [5]. In 2006, NPS simultaneously sampled lichen communities and remeasured elemental concentrations in a new sampling grid using five strata based on distance from the haul road [4, 18]. The 2001 and 2006 analyses were formally included as a protocol of the NPS Arctic Network Inventory and Monitoring Program, ensuring decadal monitoring over the long term [19]. This protocol strives to determine the status and trends of key ecological indicators using a stable sampling design, methodology, and analysis. In 2017, NPS conducted the third measurement of moss elemental concentration in CAKR, co-located with the second sampling of lichen communities.

This paper presents the results of a decadal spatial analysis of contaminant deposition based on the resampling of locations originally sampled in 2006 [4] and fitting the 2006 and 2017 data to the model used previously for the 2001 and 2006 data. Our objectives were to: 1) describe the results of the spatial analysis of 2017 data using a Bayesian geostatistical model of elemental concentrations of Zn, Pb, and Cd, 2) summarize strata-based predictions of elemental concentrations, 3) summarize and map pointwise posterior predictions for elemental concentrations of these 3 metals, and 4) assess trends in elemental concentrations between 2001–2017 by determining the probability of decrease by element for both points and strata.

This article describes the results of a long-term monitoring protocol, and as such, it represents the latest findings in a series of research articles using stable methodology and analyses. For brevity, we recommend readers refer to earlier articles [4, 5, 18] for greater detail on study design, methods, lab analyses, and statistics.

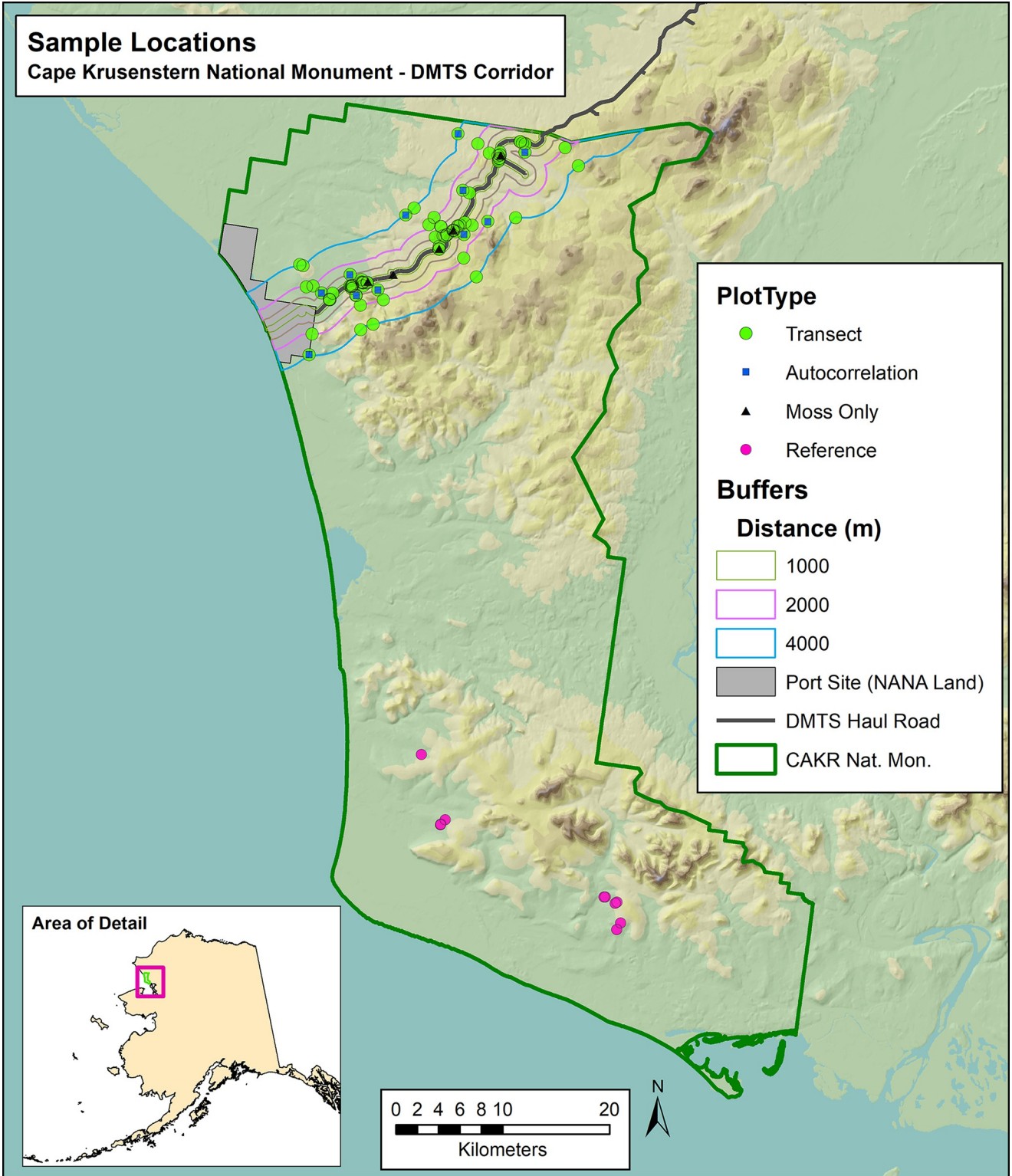

**Fig 1. Study area and location of sample plots, Cape Krusenstern National Monument and the Delong Mountain Transportation System haul road, Alaska.**

## Methods

### Ethics statement

This study was approved by the National Park Service-Western Arctic National Parklands interdisciplinary project review team under the provisions of the National Environmental Policy Act [20]. The project was assessed to be low in impacts to natural resources, cultural resources and subsistence and was granted a categorical exclusion from further review.

### Study area, field and lab methods

Methods for this long-term monitoring protocol are thoroughly described in a previous open-access publication (Neitlich et al. [4]) and in Swanson and Neitlich [19] including study area, study design, field sampling, and lab analyses. In brief, the goal of this study was to detect trends in Zn, Pb, and Cd concentrations in the feather moss *Hylocomium splendens* in CAKR adjacent to the DMTS haul road between 2006 and 2017. To obtain a landscape-scale assessment of contaminants, samples of the *H. splendens* were obtained in 2017 immediately adjacent to the 104 long-term vegetation monitoring plots established in 2006 [4, 18] and from 6 other roadside collection points (Fig 1). These 104 vegetation monitoring plots formed the network from which ≥95% of the samples in both 2006 and 2017 were obtained.

Earlier studies along the DMTS haul road showed that heavy metal deposition declined in a roughly logarithmic curve from the road [5, 6]. Sampling distances from the road were thus chosen in 2006 to obtain an even spread of values along this deposition gradient. Twelve non-linear "transects" were created according to the distances in Table 1, with points at 10, 50, 100, 300, 1000, 2000, and 4000 m. To help account for spatial autocorrelation in the model, one additional plot (termed an "autocorrelation plot") was established in each transect at a random distance 10 to 20 m from the 1,000, 2,000, or 4,000 m plot in each transect. Each "transect" consisted of a group of points within one of the two most common land cover classes in the study area at specified distance classes. Points were chosen at random in GIS in close geographic proximity to a line, but did not represent a strictly linear feature. Final locations were additionally randomized using a random number generator to select a group of at least 10 contiguous pixels of the target landcover class closest to the transect line at a particular distance. Linear transects were tested in

**Table 1. Number of plots and tissue samples by distance from the DMTS haul road, 2017.** Standard and autocorrelation plots sampled both vegetation and moss tissue. Autocorrelation plots were established to provide autocorrelation estimation in the posterior predictions model. A tissue sample, and in some cases a field duplicate sample, was collected at each plot with the exception of four plots (distances 10, 300, 1000, and 4000 m) at in which no tissue was available. The six samples closest to the road did not co-occur with lichen plots. All of the 131 samples from 104 plots and 6 additional tissue sample locations in 2017 were used in the current model.

| Distance from Road (m) | Plots (N) | Plot Type | Tissue Samples (N) |
|---|---|---|---|
| 3 | 0 | Moss Tissue Only | 6 |
| 10 | 12 | Standard | 12 |
| 50 | 12 | Standard | 15 |
| 100 | 12 | Standard | 17 |
| 300 | 12 | Standard | 14 |
| 1,000 | 12 | Standard | 14 |
| 1,000 | 8 | Autocorrelation | 8 |
| 2,000 | 11 | Standard | 14 |
| 2,000 | 1 | Autocorrelation | 1 |
| 4,000 | 11 | Standard | 15 |
| 4,000 | 3 | Autocorrelation | 2 |
| 42,000–48,000 | 4 | Reference 1 | 5 |
| 60,000–64,200 | 6 | Reference 2 | 7 |

the planning stages, but rejected because it was not possible to find both the correct land cover classes and distances along a straight line. The two most common landcover types accounting for more than 66% of the study area were Upland Moist Dwarf Birch-Ericaceous Shrub and Upland Moist Dwarf Birch-Tussock Shrub [21]. These closely-related classes were each dominated by cottongrass (*Eriophorum vaginatum*), dwarf bich (*Betula nana*) and mixed Ericaceous shrubs (e.g., *Ledum decumbens*, *Vaccinium uliginosum*, *Vaccinium vitis-idaea*).

Ten reference samples were obtained in southern CAKR at distances of 42,000 to 64,000 m from the DMTS haul road adjacent to the long-term vegetation plots there. To avoid the presence of high dust and mud immediately adjacent to the road, long term vegetation monitoring plots transects started at a distance of 10 m from the road. To capture the higher contaminant values immediately adjacent to the road, therefore, a small number of roadside moss samples unassociated with the long-term monitoring plots were also obtained in both 2006 and 2017. A slightly larger number of samples (131) were collected in 2017 (Table 1) than in 2006 (125). This was due to a greater number of field duplicate samples and two additional stand-alone roadside moss samples. Moss samples were either collected and stored dry in heavy duty chemical sample bags or collected wet and frozen until cleaning and analysis, and were collected and stored double bagged using sterile techniques [4]. The concentrations reported from lab analysis are mg/kg of analyte in *Hylocomium splendens* moss tissue, dry weight.

## Posterior predictions modeling

We analyzed the 2006 and 2017 chemical concentration data using a geostatistical model that accounts for spatial correlation among observations. We let $y_{ij}$ denote the observed chemical concentration at spatial location $i = 1,\ldots,n$ for measurement $j = 1,\ldots,J_i$ and $\mathbf{x}_i$ denote a vector of predictor variables (including 1 as the first element corresponding to an intercept parameter) measured at location $i$. Our analysis included distance to road (natural logarithmic scale, referred to as "log"), an indicator for south side of the road, and their interaction as predictor variables. We formulated our geostatistical model as

$$\log(y_{ij}) = \mathbf{x}'_i\boldsymbol{\beta} + \alpha_i + \eta_i + \epsilon_{ij}, \qquad (1)$$

where $\boldsymbol{\beta}$ is a vector of regression coefficients. The $\alpha_i$ terms represent site-level random effects that account for the repeated measurements at sampled locations. We assumed these terms were independent across sites and were normally distributed with mean zero and variance $\sigma_1^2$. To account for spatial correlation among observations, we also included a spatial random effect $\eta_i$ for each location. Defining $\boldsymbol{\eta} \equiv (\eta_1, \eta_2,\ldots,\eta_n)'$, we assumed $\boldsymbol{\eta} \sim N(\mathbf{0}, \Sigma)$ where the covariance among spatial effects was specified as a function of the distance between the corresponding locations. Specifically, we used the exponential covariance function so that

$$\text{cov}(\eta_i, \eta_{\tilde{i}}) = \sigma_2^2 \exp(-d_{i\tilde{i}}/\theta), \qquad (2)$$

where $\sigma_2^2$ and $\theta$ are parameters to be estimated and $d_{i\tilde{i}}$ is the distance between locations $i$ and $\tilde{i}$. The measurement errors $\epsilon_{ij}$ were assumed to be independent and normally distributed with mean zero and variance $\sigma_0^2$. Separate geostatistical models were fit for each combination of element (Cd, Pb, and Zn) and year (2006 and 2017).

We used Bayesian methods to fit these models and specified weakly informative prior distributions for each parameter. The prior distribution for $\boldsymbol{\beta}$ was specified as multivariate normal with mean equal to a zero vector and covariance matrix $100\mathbf{I}$. For $\sigma_0^2$, $\sigma_1^2$, and $\sigma_2^2$ we assumed Gamma(2,2) (shape and rate parameterization) priors. We specified a Half-normal(0,5000) prior distribution for the spatial range parameter $\theta$ based on the scale of the observed distances between sampled locations. Note that these prior distributions differed slightly from those

assumed by Neitlich et al. [4]. However, for the 2006 data, inferences were not sensitive to the change in prior distributions based on comparisons to previous results (e.g., see tables in Results section). All statistical analyses were conducted in R [22]. Each model was fit with Markov chain Monte Carlo (MCMC) using Stan [23] called from R with the rstan package [24]. Our Bayesian geostatistical models were fit using 4 chains of 4,000 iterations each where the first half (2,000 iterations) of each chain was discarded as adaptation/burn-in and the second half was used for inference.

Posterior predictions of chemical concentrations were made across a grid of 2,374 total locations for 200 random posterior draws (to reduce computation time). The prediction grid was established by Neitlich et al. [4] and consists of five strata that were defined based on proximity of points to the road. For posterior draw $k$, let $\hat{y}_{\text{year},l}^{(k)}$ denote the predicted concentration (on the original scale) for a particular year and prediction grid point $l$. We created maps summarizing the posterior predictive distribution at each grid point based on the posterior mean, 2.5% posterior quantile, and 97.5% posterior quantile. Additional maps were created to summarize the posterior distribution of percent change and the posterior probability of decrease at each prediction grid point. Percent change associated with posterior draw $k$ and point $l$ was calculated as

$$100 \frac{(\hat{y}_{2017,l}^{(k)} - \hat{y}_{2006,l}^{(k)})}{\hat{y}_{2006,l}^{(k)}}. \qquad (3)$$

The posterior probability of decrease was calculated based on the proportion of posterior draws where $\hat{y}_{2006,l}^{(k)}$ was larger than $\hat{y}_{2017,l}^{(k)}$.

Additional summaries of the posterior predictive distributions were made by strata. Following Neitlich et al. [4], we defined the average concentration for strata $t$ and posterior draw $k$ as

$$\hat{b}_{\text{year},t}^{(k)} = m_t^{-1} \sum_{l \in \text{strata } t} \hat{y}_{\text{year},l}^{(k)} \qquad (4)$$

where $m_t$ is the total number of prediction grid points in strata $t$. We then summarized the posterior predictive distributions for the change in average concentration, the percent change in average concentration, and probability of decrease in average concentration. Change in average concentration was calculated as $\hat{b}_{2017,t}^{(k)} - \hat{b}_{2006,t}^{(k)}$ and percent change in average concentration was calculated as $100(\hat{b}_{2017,t}^{(k)} - \hat{b}_{2006,t}^{(k)})/\hat{b}_{2006,t}^{(k)}$. The posterior probability of decrease in average concentration was based on the proportion of posterior draws where $\hat{b}_{2006,t}^{(k)}$ was larger than $\hat{b}_{2017,t}^{(k)}$.

## Results and discussion

### Spatial model

Summaries of the posterior distributions for the regression coefficients are shown in Table 2. Based on these results, we can infer that log-concentrations decrease as log-distance to the haul road increases for all three elements and both years because the credible intervals for $\beta_1$ include only negative values. For Cd, the increase in the intercept parameter from 2006–2017 suggests that log-concentrations increased at locations close to the road. However, the posterior mean of the road distance coefficient decreased from 2006–2017, indicating that Cd log concentrations decreased more rapidly as distance from road increases. The posterior means of regression coefficients for Pb in 2017 show that log concentrations are estimated to be lower closer to the road but decrease less rapidly as distance from road increased relative to 2017.

**Table 2. Summaries of posterior distributions for regression coefficients in Eq 1.** The means and standard error of the posterior distributions are provided along with the lower 2.5% and upper 97.5% endpoints of the 95% credible interval.

| | Cd | | | | | | | |
|---|---|---|---|---|---|---|---|---|
| | 2006 | | | | 2017 | | | |
| Parameter | Mean | SE | Lower | Upper | Mean | SE | Lower | Upper |
| $\beta_0$ | 3.778 | 0.296 | 3.226 | 4.384 | 4.626 | 0.467 | 3.734 | 5.598 |
| $\beta_1$ | -0.504 | 0.032 | -0.567 | -0.438 | -0.618 | 0.045 | -0.705 | -0.529 |
| $\beta_2$ | -0.471 | 0.233 | -0.927 | -0.010 | -0.466 | 0.304 | -1.049 | 0.143 |
| $\beta_3$ | 0.014 | 0.044 | -0.076 | 0.097 | -0.037 | 0.065 | -0.166 | 0.087 |
| | Pb | | | | | | | |
| | 2006 | | | | 2017 | | | |
| Parameter | Mean | SE | Lower | Upper | Mean | SE | Lower | Upper |
| $\beta_0$ | 7.849 | 0.389 | 7.121 | 8.641 | 6.775 | 0.354 | 6.078 | 7.478 |
| $\beta_1$ | -0.629 | 0.044 | -0.718 | -0.544 | -0.497 | 0.041 | -0.576 | -0.416 |
| $\beta_2$ | -0.498 | 0.300 | -1.096 | 0.100 | -0.239 | 0.271 | -0.764 | 0.315 |
| $\beta_3$ | -0.025 | 0.058 | -0.138 | 0.092 | -0.069 | 0.054 | -0.177 | 0.036 |
| | Zn | | | | | | | |
| | 2006 | | | | 2017 | | | |
| Parameter | Mean | SE | Lower | Upper | Mean | SE | Lower | Upper |
| $\beta_0$ | 9.165 | 0.404 | 8.414 | 10.012 | 9.016 | 0.423 | 8.205 | 9.872 |
| $\beta_1$ | -0.535 | 0.037 | -0.608 | -0.463 | -0.533 | 0.041 | -0.612 | -0.453 |
| $\beta_2$ | -0.766 | 0.260 | -1.282 | -0.260 | -0.626 | 0.296 | -1.203 | -0.050 |
| $\beta_3$ | 0.059 | 0.053 | -0.043 | 0.161 | 0.029 | 0.060 | -0.088 | 0.147 |

The posterior distributions of the regression coefficients for Zn are similar for 2006 and 2017. To better explore the differences in estimated log concentrations across years, we summarize the posterior predictive distributions for these models in Table 2.

## Spatial patterns of deposition

**Strata-based summarization.** The strata used in this study followed Table 3 and Fig 3 in Neitlich et al. [4] and are based on distance from the DMTS haul road as follows: Stratum 1, 0–100 m; Stratum 2, 100–2,000 m; Stratum 3, 2,000–4,000 m; Stratum 4, 4,000–5,000 m, Stratum 5, 30,000–64,000 m. Each stratum represents a distance on both the north and south side of the road with the exception of Stratum 5, which occurs only on the south side of the road. Strata-based summarization is presented for Zn, Pb, and Cd independently according to side of road as well as grouped.

In Stratum 1, change in Zn concentrations in *Hylocomium splendens* moss between 2006–2017 was highly uncertain with a 56% probability of decrease (Table 3). In Strata 2, 3, and 4, the probabilities of a decrease in Zn between 2006–2017 were >90% with mean decreases of 12–15 mg/kg (representing a percent decrease between 11–20%). In Stratum 5, there was a 78% probability of a decrease from 40 to 31 mg/kg. The main difference between the 2001–2006 period and that of 2006–2017 is that Zn concentrations did not decrease in 2017 in Stratum 1 as they did for every other stratum in 2017 (Fig 2).

Changes in Pb levels between 2006–2017 were considerably different from that of Zn (Table 4). There was 100% probability of a decrease of 80 mg/kg (from 198 to 118 mg/kg) in Stratum 1. Other strata had only moderate probabilities of decrease (29–82%) with very small change quantities (0–1.7 mg/kg). These findings do not point to any meaningful change in Pb levels over this time period at distances >100 m, but a very sizeable decrease close to the

**Table 3. Modeled concentrations (mg/kg dry weight) of zinc (Zn) in *Hylocomium splendens* moss tissue in 2006 and 2017, average change, percent change, and probability of decrease for 5 strata.** Results were tabulated for the whole data set as well as separately for the north and south sides of the road. Results are presented for the posterior mean, standard deviation (SD), and the lower and upper bounds representing the 2.5% and 97.5% endpoints of the 95% credible interval. Pr. Dec. presents the posterior probability of decrease in concentration between 2006 and 2017.

| Zn | | | | | | | | | | | | | | | | | |
|---|---|---|---|---|---|---|---|---|---|---|---|---|---|---|---|---|---|
| **All** | | **2006** | | | | **2017** | | | | **Average Change** | | | | **Percent Change** | | | | **Pr. Dec.** |
| Stratum | N | Mean | SD | Lower | Upper | Mean | SD | Lower | Upper | Mean | SD | Lower | Upper | Mean | SD | Lower | Upper | % |
| 1 | 697 | **549.9** | 47.5 | 471.3 | 661.6 | **531.3** | 49.6 | 445.0 | 643.4 | **-18.6** | 63.8 | -153.7 | 95.7 | **-2.8** | 11.3 | -25.1 | 19.4 | **56** |
| 2 | 743 | **101.4** | 6.2 | 89.9 | 114.8 | **89.6** | 5.4 | 80.4 | 100.5 | **-11.8** | 8.3 | -30.0 | 5.4 | **-11.3** | 7.8 | -26.8 | 5.8 | **92** |
| 3 | 507 | **69.0** | 5.7 | 58.1 | 80.4 | **54.6** | 4.3 | 46.8 | 63.2 | **-14.4** | 6.7 | -27.7 | -0.9 | **-20.4** | 8.6 | -35.7 | -1.7 | **97** |
| 4 | 121 | **67.9** | 8.7 | 53.5 | 90.7 | **53.4** | 6.5 | 42.8 | 65.9 | **-14.5** | 10.5 | -34.7 | 7.1 | **-20.2** | 13.7 | -43.0 | 13.0 | **94** |
| 5 | 289 | **40.2** | 11.2 | 24.0 | 68.5 | **30.7** | 7.5 | 19.2 | 49.6 | **-9.5** | 13.7 | -41.7 | 11.8 | **-18.2** | 28.0 | -63.7 | 37.2 | **78** |
| **North** | | **2006** | | | | **2017** | | | | **Average Change** | | | | **Percent Change** | | | | **Pr. Dec.** |
| Stratum | N | Mean | SD | Lower | Upper | Mean | SD | Lower | Upper | Mean | SD | Lower | Upper | Mean | SD | Lower | Upper | % |
| 1 | 351 | **706.4** | 80.1 | 579.8 | 865.0 | **672.2** | 84.0 | 534.8 | 859.1 | **-34.2** | 110.7 | -277.5 | 161.0 | **-3.8** | 15.1 | -31.9 | 26.9 | **58** |
| 2 | 360 | **121.9** | 10.4 | 101.3 | 141.4 | **108.6** | 8.6 | 94.2 | 127.1 | **-13.4** | 13.5 | -37.7 | 13.8 | **-10.3** | 10.7 | -27.8 | 11.4 | **83** |
| 3 | 215 | **75.0** | 10.0 | 58.8 | 95.0 | **57.5** | 6.9 | 45.4 | 73.2 | **-17.5** | 11.7 | -39.4 | 4.3 | **-22.1** | 13.6 | -44.4 | 7.2 | **95** |
| 4 | 46 | **67.2** | 10.8 | 47.0 | 88.1 | **51.6** | 8.7 | 37.8 | 71.8 | **-15.6** | 14.5 | -42.9 | 11.8 | **-21.0** | 20.1 | -48.8 | 22.0 | **88** |
| **South** | | **2006** | | | | **2017** | | | | **Average Change** | | | | **Percent Change** | | | | **Pr. Dec.** |
| Stratum | N | Mean | SD | Lower | Upper | Mean | SD | Lower | Upper | Mean | SD | Lower | Upper | Mean | SD | Lower | Upper | % |
| 1 | 346 | **391.1** | 46.2 | 316.8 | 492.6 | **388.2** | 48.3 | 313.4 | 483.3 | **-2.9** | 65.9 | -132.0 | 106.5 | **0.5** | 16.4 | -30.6 | 28.5 | **46** |
| 2 | 383 | **82.1** | 6.8 | 69.6 | 95.6 | **71.8** | 6.0 | 60.6 | 85.0 | **-10.3** | 9.7 | -28.4 | 7.4 | **-11.8** | 11.2 | -30.2 | 10.5 | **83** |
| 3 | 292 | **64.5** | 6.2 | 53.3 | 77.5 | **52.4** | 5.4 | 44.0 | 66.5 | **-12.1** | 8.2 | -28.8 | 3.6 | **-18.1** | 11.4 | -36.8 | 5.7 | **92** |
| 4 | 75 | **68.3** | 11.2 | 51.2 | 95.1 | **54.5** | 8.6 | 41.0 | 71.1 | **-13.8** | 13.3 | -42.1 | 9.0 | **-18.5** | 16.9 | -47.4 | 15.4 | **88** |
| 5 | 289 | **40.2** | 11.2 | 24.0 | 68.5 | **30.7** | 7.5 | 19.2 | 49.6 | **-9.5** | 13.7 | -41.7 | 11.8 | **-18.2** | 28.0 | -63.7 | 37.2 | **78** |

DMTS haul road. The lack of a high probability of decreases in Strata 2–4 between 2006–2017 contrasts to the 100% probability of decreases in these strata between 2001–2006 [4] (Fig 2).

The change in Cd during this time period was yet again different from that of Zn and Pb (Table 5). In Stratum 1, there was an 80% probability of a decrease of 0.60 mg/kg from 5.64 mg/kg to 4.64 mg/kg. In all other strata, there was a 97–100% probability of decrease, with percent decreases ranging from 41–51%. This level of change represents a much more sizeable decrease than was recorded between 2001–2006 [4], during which meaningful change occurred in Stratum 1 but was either small in quantity or a possible increase in more distant strata (Fig 2).

**Point-based mapping.** Point-based mapping provides an opportunity to examine spatial patterns in finer detail than strata-based mapping. In contrast to the widespread and substantial decrease in concentrations of Zn, Pb, and Cd in moss tissue from 2001–2006, the period from 2006–2017 was more variable. For example, while a decrease in Zn of ≥25% was predominant in the DMTS corridor from 2001–2006 (Fig 8 in [4]), the predominant changes in Zn ranged from +50% to -50% between 2006–2017 (Figs 3–5). Similarly, the posterior probability of decrease was much higher over the corridor as a whole in the earlier monitoring period. Between 2001–2006, a dominant portion of the DMTS corridor out to 2,000 m showed ≥75% probability of decrease (Fig 11 in [4]), compared to a patchy mosaic of 40–80% probability of decrease between 2006–2017 (Fig 6). On a landscape level, the largest change in Zn concentrations between 2006–2017 was the decrease from the 60–100 mg/kg class to the 13–60 mg/kg class over a large portion of the north side of the road (Figs 4 and 5). This decrease is significant as the provisional threshold for effects on lichen species richness is approximately 58–70 mg/kg of Zn in *Hylocomium* moss tissue, meaning this area may once again fall below levels

## Metal concentrations by year and stratum

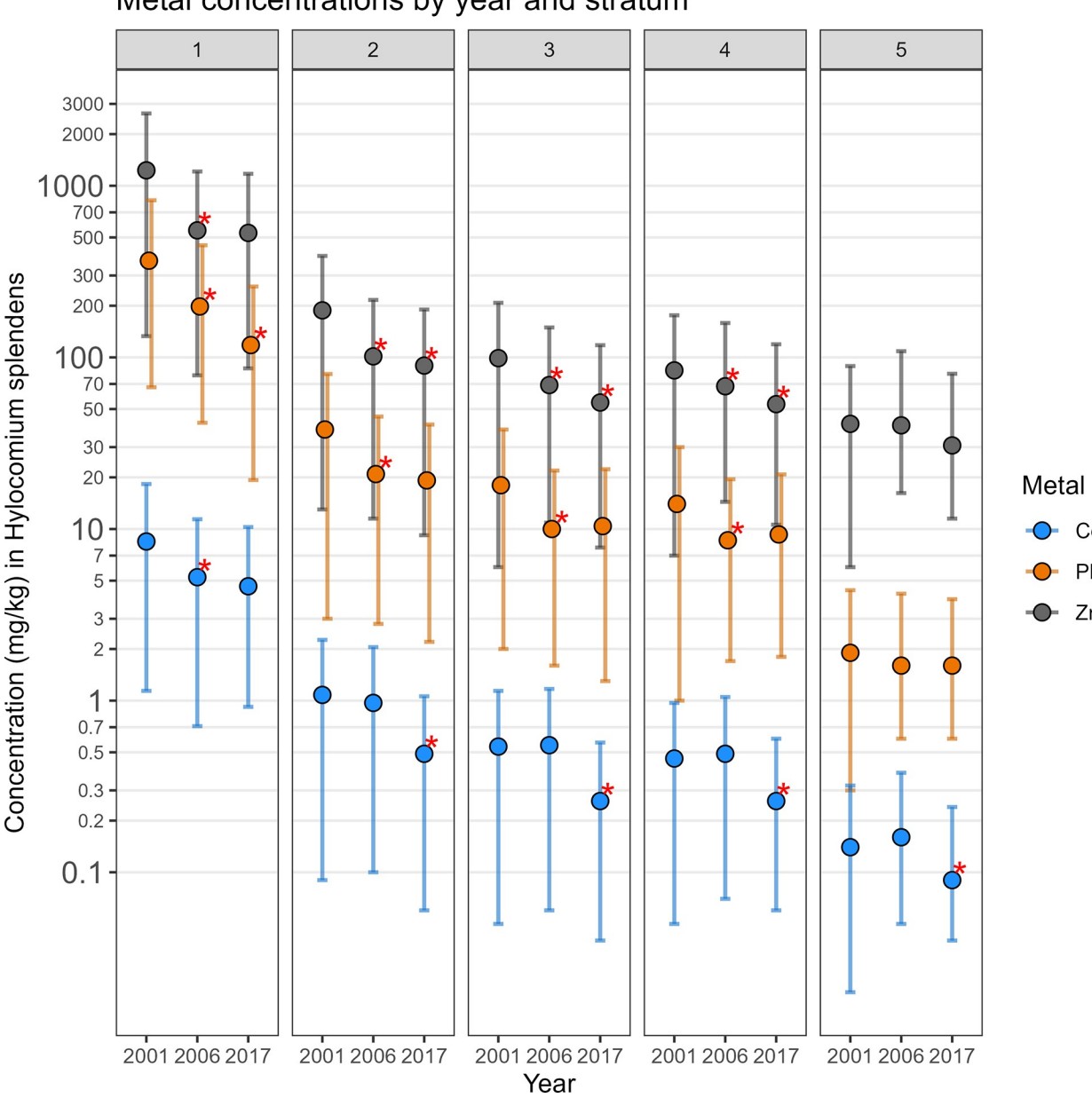

**Fig 2. Changes in mean modeled Zn, Pb, and Cd concentrations in *Hylocomium splendens* between 2001 and 2017.** Posterior means that showed >90% probability of decrease from one sampling period to the next are marked with a red asterisk. Error bars show the high and low bounds of the 95% credible interval. The 2001–2006 change data are from Neitlich et al. [4] using exactly the same modeling procedures as the 2006–2017 analysis.

that cause lichen mortality [18]. Less clear is whether recovery can occur at these levels given both legacy Zn in soils [25] and ongoing lower levels of Zn deposition. The areas closest to the haul road (≤100 m) showed the persistence of Zn concentrations in the 400–1200 mg/kg range and a predominance of increase of up to 25%. The widespread probabilities of decrease between 40–80% pointed to a lack of clear trends in this zone of most elevated concentrations. This is corroborated by the 56% probability of increase in Stratum 1 (Table 3).

**Table 4. Modeled concentrations (mg/kg dry weight) of lead (Pb) in *Hylocomium splendens* moss tissue in 2006 and 2017, average change, percent change, and probability of decrease for 5 strata.** Results were tabulated for the whole data set as well as separately for the north and south sides of the road. Results are presented for the posterior mean, standard deviation (SD), and the lower and upper bounds representing the 2.5% and 97.5% endpoints of the 95% credible interval. Pr. Dec. presents the posterior probability of decrease in concentration between 2006 and 2017.

| Pb | | | | | | | | | | | | | | | | | |
|---|---|---|---|---|---|---|---|---|---|---|---|---|---|---|---|---|---|
| **All** | | **2006** | | | **2017** | | | | **Average Change** | | | | **Percent Change** | | | | **Pr. Dec.** |
| Stratum | N | Mean | SD | Lower | Upper | Mean | SD | Lower | Upper | Mean | SD | Lower | Upper | Mean | SD | Lower | Upper | % |
| 1 | 697 | **198.4** | 23.6 | 156.4 | 252.1 | **118.2** | 10.4 | 98.7 | 140.8 | **-80.2** | 26.2 | -142.6 | -37.3 | **-39.6** | 9.0 | -55.9 | -23.0 | **100** |
| 2 | 743 | **20.9** | 1.6 | 18.1 | 24.3 | **19.2** | 1.0 | 17.0 | 21.4 | **-1.7** | 1.8 | -5.7 | 1.6 | **-7.7** | 8.1 | -23.7 | 8.6 | **82** |
| 3 | 507 | **10.0** | 0.9 | 8.4 | 11.9 | **10.4** | 0.7 | 9.1 | 11.9 | **0.4** | 1.2 | -2.1 | 2.6 | **5.1** | 11.9 | -17.4 | 29.5 | **38** |
| 4 | 121 | **8.6** | 1.1 | 6.9 | 10.9 | **9.3** | 1.0 | 7.5 | 11.5 | **0.7** | 1.4 | -2.1 | 3.2 | **9.8** | 17.1 | -19.7 | 40.5 | **29** |
| 5 | 289 | **1.6** | 0.4 | 1.0 | 2.6 | **1.6** | 0.3 | 1.0 | 2.3 | **0.0** | 0.6 | -1.1 | 1.0 | **5.2** | 35.4 | -48.9 | 85.1 | **51** |
| **North** | | **2006** | | | **2017** | | | | **Average Change** | | | | **Percent Change** | | | | **Pr. Dec.** |
| Stratum | N | Mean | SD | Lower | Upper | Mean | SD | Lower | Upper | Mean | SD | Lower | Upper | Mean | SD | Lower | Upper | % |
| 1 | 351 | **248.5** | 37.4 | 179.2 | 335.5 | **141.4** | 17.8 | 109.7 | 176.3 | **-107.2** | 42.3 | -212.9 | -39.4 | **-41.8** | 11.4 | -63.0 | -19.3 | **99** |
| 2 | 360 | **27.7** | 2.6 | 23.4 | 33.4 | **24.8** | 1.7 | 21.6 | 27.9 | **-2.9** | 3.1 | -8.2 | 2.1 | **-9.8** | 10.4 | -27.1 | 9.0 | **82** |
| 3 | 215 | **13.1** | 1.8 | 10.1 | 17.4 | **12.0** | 1.3 | 9.7 | 14.8 | **-1.1** | 2.2 | -6.0 | 2.8 | **-7.1** | 15.7 | -34.3 | 25.0 | **73** |
| 4 | 46 | **11.1** | 2.2 | 7.7 | 15.3 | **10.1** | 1.6 | 7.5 | 13.4 | **-1.0** | 2.6 | -5.9 | 4.1 | **-5.8** | 22.1 | -42.4 | 45.0 | **64** |
| **South** | | **2006** | | | **2017** | | | | **Average Change** | | | | **Percent Change** | | | | **Pr. Dec.** |
| Stratum | N | Mean | SD | Lower | Upper | Mean | SD | Lower | Upper | Mean | SD | Lower | Upper | Mean | SD | Lower | Upper | % |
| 1 | 346 | **147.6** | 22.9 | 111.4 | 197.0 | **94.7** | 11.4 | 74.6 | 123.8 | **-52.9** | 25.7 | -104.6 | -3.9 | **-34.3** | 12.9 | -55.4 | -2.9 | **98** |
| 2 | 383 | **14.6** | 1.5 | 11.9 | 17.7 | **14.0** | 1.1 | 11.9 | 16.2 | **-0.6** | 1.8 | -4.8 | 2.9 | **-3.1** | 12.4 | -26.8 | 23.1 | **62** |
| 3 | 292 | **7.7** | 0.8 | 6.2 | 9.6 | **9.2** | 0.8 | 7.7 | 11.2 | **1.6** | 1.2 | -0.9 | 3.8 | **22.2** | 17.3 | -10.0 | 59.9 | **10** |
| 4 | 75 | **7.1** | 1.1 | 5.2 | 9.2 | **8.9** | 1.3 | 6.8 | 11.6 | **1.8** | 1.7 | -1.7 | 4.9 | **27.4** | 26.2 | -20.5 | 83.3 | **12** |
| 5 | 289 | **1.6** | 0.4 | 1.0 | 2.6 | **1.6** | 0.3 | 1.0 | 2.3 | **0.0** | 0.6 | -1.1 | 1.0 | **5.2** | 35.4 | -48.9 | 85.1 | **51** |

The concentrations of Pb in moss tissue between 2006–2017 changed in a bimodal pattern. The area immediately adjacent to the road decreased predominantly by ≥-25%, while more distant areas north of the road tended toward increases of ≤50%. Large portions south of the road increased by 25–100%. (Figs 7–9). The probabilities of decrease (Fig 10) were strongest immediately adjacent to the DMTS haul road (predominantly 80–100%), mixed on the north side of the road (20–80%) and predominantly low on the south side of the road (20–40%, i.e., 60–80% chance of increase). The high probability of decreases adjacent to the road (predominated by the 0–25 and 25–50% classes) mirrors the 100 percent probability of a decrease of 39% (or approximately 80 mg/kg) in Table 4. The probability of decrease within the 1,000 m haul road buffer zone was predominantly in the 40–60% range with low overall percent change values (0–25%), indicating little change. Pb concentrations stayed at ≥10 mg/kg in moss tissue in this zone and commonly in the 200–3,200 mg/kg range adjacent to the haul road. There appeared to be a 60–80 percent probability of decrease in CAKR adjacent to (and especially on) the north side of the Port Site, with decreases of up to 25%. The high dominance of the 20–40% probability of decrease (i.e., 60–80% probability of increase) throughout the south side of the corridor at distances >1,000 m from the road is suggestive of an increase, though certainly not definitive. Table 4 provides insight that regardless of the probabilities, the mean change in these more distant southern areas is quite small (<2 mg/kg Pb in moss tissue).

As also seen in Fig 2 and Table 5, change in Cd was stronger and more certain at distances beyond the 100 m of the immediate DMTS haul road corridor (Figs 11–14). Almost the entire study area outside of the immediate corridor decreased by 25–75% with about equal amounts in the 25–50% and 50–75% classes. The probabilities of decrease were predominantly in the >95% and the 80–95% classes. Along the road, trends were not as clear, as there was at the

**Table 5. Modeled concentrations (mg/kg dry weight) of cadmium (Cd) in *Hylocomium splendens* moss tissue in 2006 and 2017, average change, percent change, and probability of decrease for 5 strata.** Results were tabulated for the whole data set as well as separately for the north and south sides of the road. Results are presented for the posterior mean, standard deviation (SD), and the lower and upper bounds representing the 2.5% and 97.5% endpoints of the 95% credible interval. Pr. Dec. presents the posterior probability of decrease in concentration between 2006 and 2017.

| Cd | | | | | | | | | | | | | | | | | |
|---|---|---|---|---|---|---|---|---|---|---|---|---|---|---|---|---|---|
| **All** | | **2006** | | | **2017** | | | | **Average Change** | | | | **Percent Change** | | | | **Pr. Dec.** |
| Stratum | N | Mean | SD | Lower | Upper | Mean | SD | Lower | Upper | Mean | SD | Lower | Upper | Mean | SD | Lower | Upper | % |
| 1 | 697 | **5.24** | 0.43 | 4.53 | 6.16 | **4.64** | 0.51 | 3.72 | 5.63 | **-0.60** | 0.69 | -1.90 | 0.83 | **-10.8** | 12.7 | -31.8 | 17.7 | **80** |
| 2 | 743 | **0.97** | 0.06 | 0.87 | 1.08 | **0.49** | 0.04 | 0.43 | 0.57 | **-0.47** | 0.07 | -0.61 | -0.34 | **-48.8** | 4.8 | -57.2 | -39.0 | **100** |
| 3 | 507 | **0.55** | 0.04 | 0.49 | 0.62 | **0.26** | 0.02 | 0.22 | 0.31 | **-0.28** | 0.05 | -0.37 | -0.21 | **-51.8** | 5.7 | -62.3 | -41.1 | **100** |
| 4 | 121 | **0.49** | 0.04 | 0.42 | 0.56 | **0.26** | 0.04 | 0.20 | 0.34 | **-0.23** | 0.05 | -0.32 | -0.13 | **-46.4** | 8.2 | -59.9 | -29.2 | **100** |
| 5 | 289 | **0.16** | 0.03 | 0.11 | 0.22 | **0.09** | 0.03 | 0.05 | 0.15 | **-0.07** | 0.04 | -0.15 | 0.02 | **-41.2** | 21.1 | -72.3 | 11.2 | **97** |

| **North** | | **2006** | | | **2017** | | | | **Average Change** | | | | **Percent Change** | | | | **Pr. Dec.** |
|---|---|---|---|---|---|---|---|---|---|---|---|---|---|---|---|---|---|---|
| Stratum | N | Mean | SD | Lower | Upper | Mean | SD | Lower | Upper | Mean | SD | Lower | Upper | Mean | SD | Lower | Upper | % |
| 1 | 351 | **6.38** | 0.70 | 5.14 | 7.67 | **5.82** | 0.87 | 4.29 | 7.57 | **-0.56** | 1.12 | -2.67 | 1.74 | **-7.7** | 17.4 | -36.2 | 30.4 | **69** |
| 2 | 360 | **1.17** | 0.09 | 1.03 | 1.35 | **0.65** | 0.06 | 0.55 | 0.80 | **-0.52** | 0.10 | -0.72 | -0.34 | **-44.0** | 6.4 | -54.0 | -31.1 | **100** |
| 3 | 215 | **0.65** | 0.06 | 0.55 | 0.77 | **0.32** | 0.05 | 0.24 | 0.41 | **-0.33** | 0.08 | -0.48 | -0.19 | **-50.2** | 8.5 | -65.9 | -32.5 | **100** |
| 4 | 46 | **0.57** | 0.08 | 0.44 | 0.74 | **0.31** | 0.07 | 0.22 | 0.47 | **-0.26** | 0.10 | -0.44 | -0.06 | **-44.0** | 14.5 | -66.1 | -13.0 | **98** |

| **South** | | **2006** | | | **2017** | | | | **Average Change** | | | | **Percent Change** | | | | **Pr. Dec.** |
|---|---|---|---|---|---|---|---|---|---|---|---|---|---|---|---|---|---|---|
| Stratum | N | Mean | SD | Lower | Upper | Mean | SD | Lower | Upper | Mean | SD | Lower | Upper | Mean | SD | Lower | Upper | % |
| 1 | 346 | **4.08** | 0.42 | 3.35 | 4.93 | **3.44** | 0.44 | 2.76 | 4.26 | **-0.64** | 0.60 | -1.68 | 0.57 | **-14.8** | 13.8 | -36.0 | 15.6 | **85** |
| 2 | 383 | **0.77** | 0.06 | 0.66 | 0.91 | **0.34** | 0.03 | 0.28 | 0.41 | **-0.43** | 0.07 | -0.59 | -0.31 | **-55.5** | 5.6 | -65.3 | -43.5 | **100** |
| 3 | 292 | **0.47** | 0.04 | 0.40 | 0.56 | **0.22** | 0.02 | 0.18 | 0.26 | **-0.25** | 0.05 | -0.35 | -0.17 | **-53.1** | 6.7 | -65.6 | -39.6 | **100** |
| 4 | 75 | **0.43** | 0.04 | 0.35 | 0.52 | **0.23** | 0.04 | 0.17 | 0.32 | **-0.21** | 0.06 | -0.32 | -0.09 | **-47.4** | 10.3 | -63.5 | -24.5 | **100** |
| 5 | 289 | **0.16** | 0.03 | 0.11 | 0.22 | **0.09** | 0.03 | 0.05 | 0.15 | **-0.07** | 0.04 | -0.15 | 0.02 | **-41.2** | 21.1 | -72.3 | 11.2 | **97** |

same time a low probability of decrease (40–80%) and percentages of increase between 0–100%. It is unclear why Cd decreased markedly during the 2006–2017 period at outer distances but only decreased in Stratum 1 between 2001–2006.

**Biological effects.** A large volume of literature describes the toxic effects of Zn deposition on lichens and bryophytes [14, 18, 26, 27]. This study documents modest levels of decrease (10–15 mg/kg in moss tissue) of Zn in areas beyond 0–100 m between 2006–2017. Neitlich et al. [18] reported a preliminary Zn threshold of 60–70 mg/kg Zn in moss tissue for effects on lichen species richness (LSR) in CAKR. In that study, Zn deposition at levels higher than this resulted in a 50–100% reduction of LSR on 11 km$^2$ and 25–100% reduction of LSR on 55 km$^2$. A total of 157 km$^2$ showed reductions in LSR of 5–100%. Di Meglio [28] reported that lichen species richness continued to decline in 2017 despite the two previous periods of decrease in Zn concentrations.

While the effects of Zn on depressing LSR and causing injury and mortality to lichens is well-known, we know much less about how lichens recover following pollution reduction. Several studies [29–33] report recolonization of impacted areas following reductions in sulfur dioxide or copper from smelters, however there are no direct analogs of the current situation in the literature. Most studies have used epiphytic lichens to study recolonization, thus have not had to consider the potential effects of residual contaminants in soil pore water on terrestrial lichens such as occur in CAKR. Melby [25] reported considerable pools of Zn, Pb, and Cd in the organic horizon pore water in the study area, and it is likely that this pore water intermittently comes in contact with lichens reaching deeper into the organic horizon (e.g., *Cladonia stygia*, *Cetraria laevigata*). Most recolonization studies concluded that reasonable recovery took decades after pollution stopped due to slow lichen growth and limitations on nearby

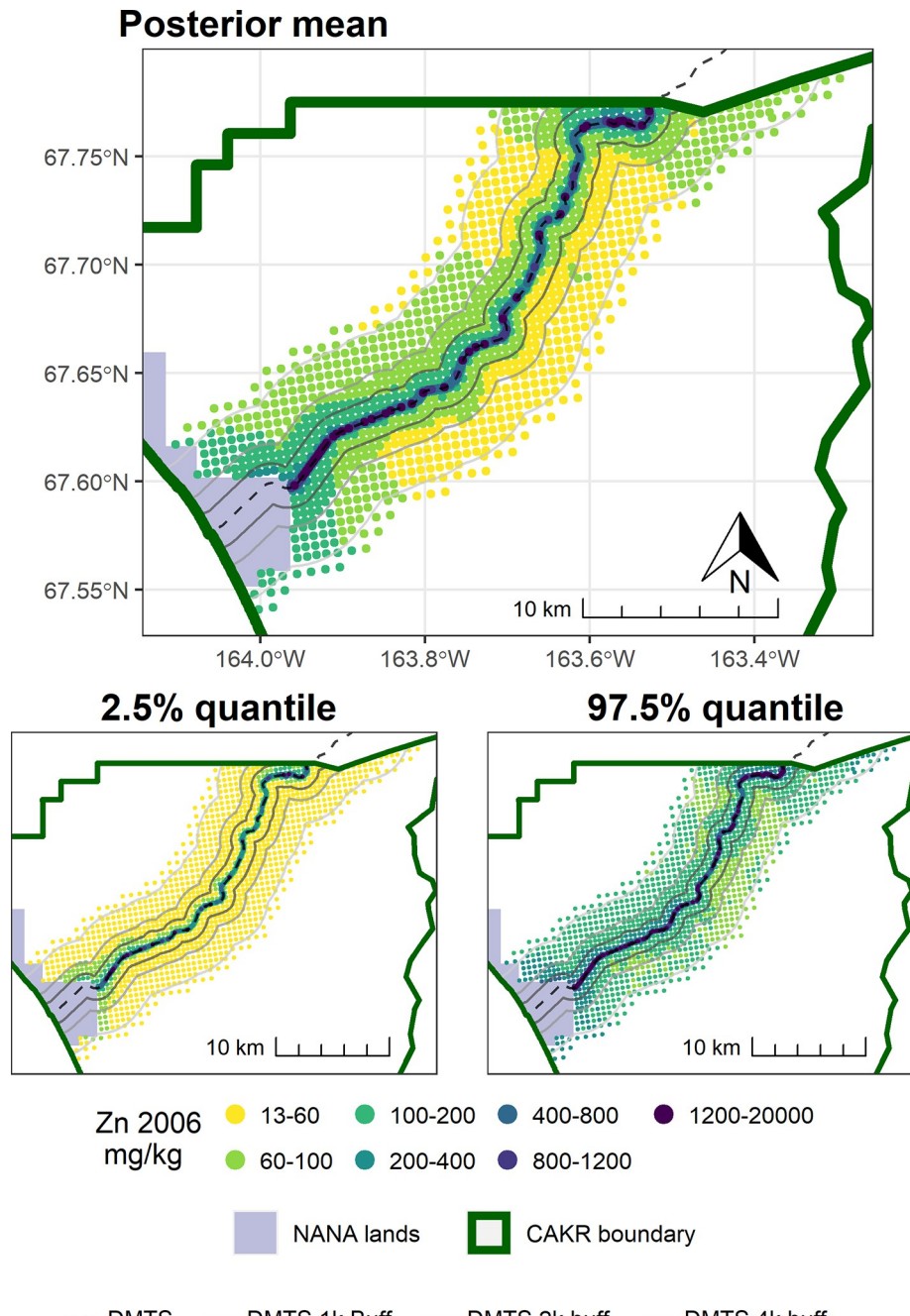

**Fig 3. Modeled 2006 Zn concentrations in moss tissue in CAKR along the DMTS haul road.** Smaller maps show the 2.5[th] and 97.5[th] percentiles (lower and upper bound of the 95% credible interval) of the modeled concentrations. DMTS haul road and distance-based buffers at 1 km, 2 km, and 4 km are shown along with NANA lands and the CAKR boundary.

propagule sources [32, 33]. Lichens and bryophytes along the DMTS haul road face reduced but ongoing pollution from both deposition and pore water exposure, which could conceivably slow recovery additionally. The Zn concentrations in 2006 that caused the decline in lichens and bryophytes along the DMTS haul road averaged between 77–566 mg at distances from 10m to 4000 m from the road. Between 2006–2017, the Zn decreases in strata for which there

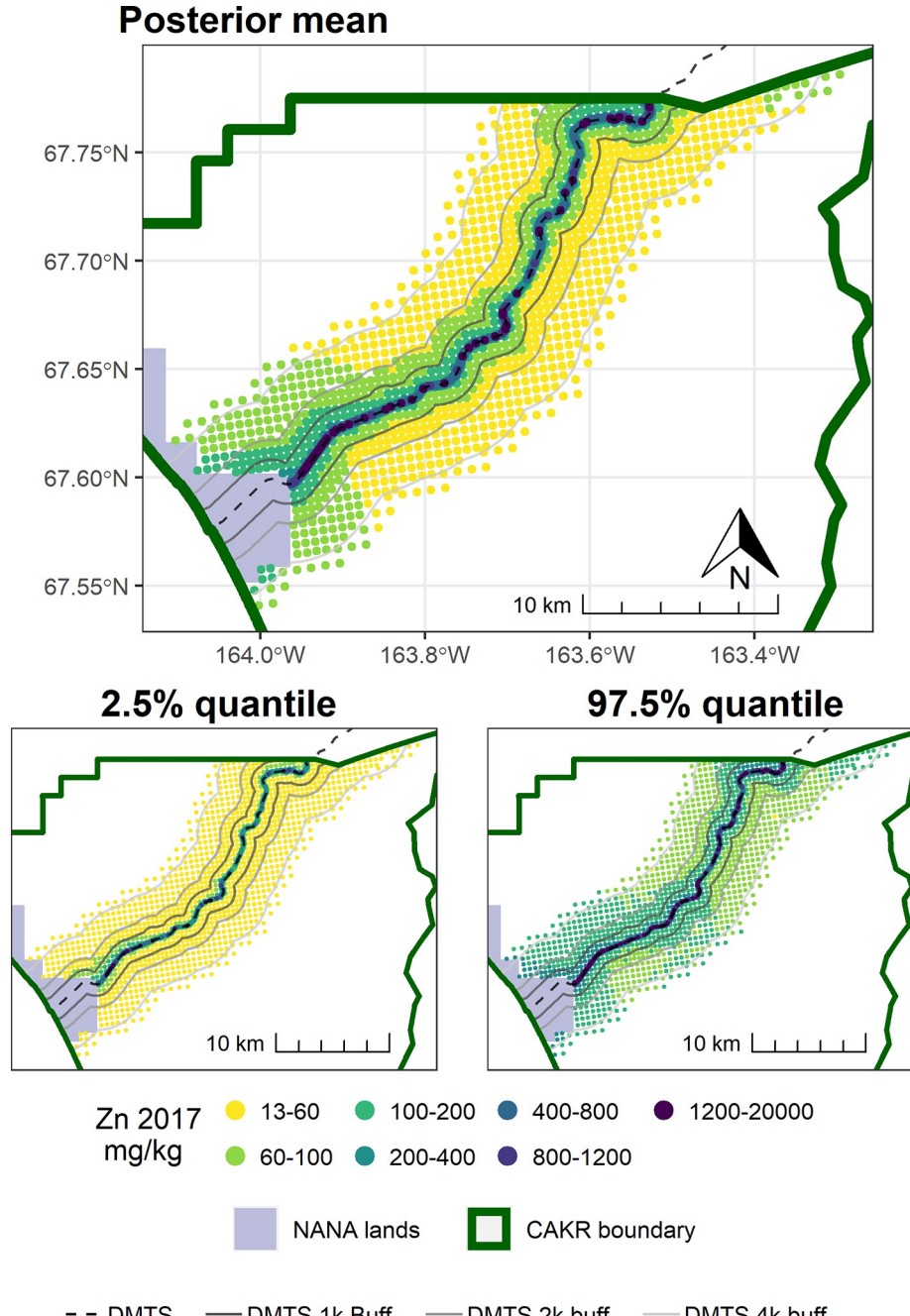

**Fig 4. Modeled 2017 Zn concentrations in moss tissue in CAKR along the DMTS haul road.** Smaller maps show the 2.5[th] and 97.5[th] percentiles (lower and upper bound of the 95% credible interval) of the modeled concentrations. DMTS haul road and distance-based buffers at 1 km, 2 km, and 4 km are shown along with NANA lands and the CAKR boundary.

was fairly high probability of decrease were ≤15mg/kg, suggesting that the bulk of the pollution signal responsible for causing the effects reported by Neitlich et al. [18] were ongoing in 2017. Still, a reduction of up to 15 mg/kg Zn could place the cleaner parts of the study area, those with higher LSR, near or below the preliminary effects threshold of 60–70 mg/kg.

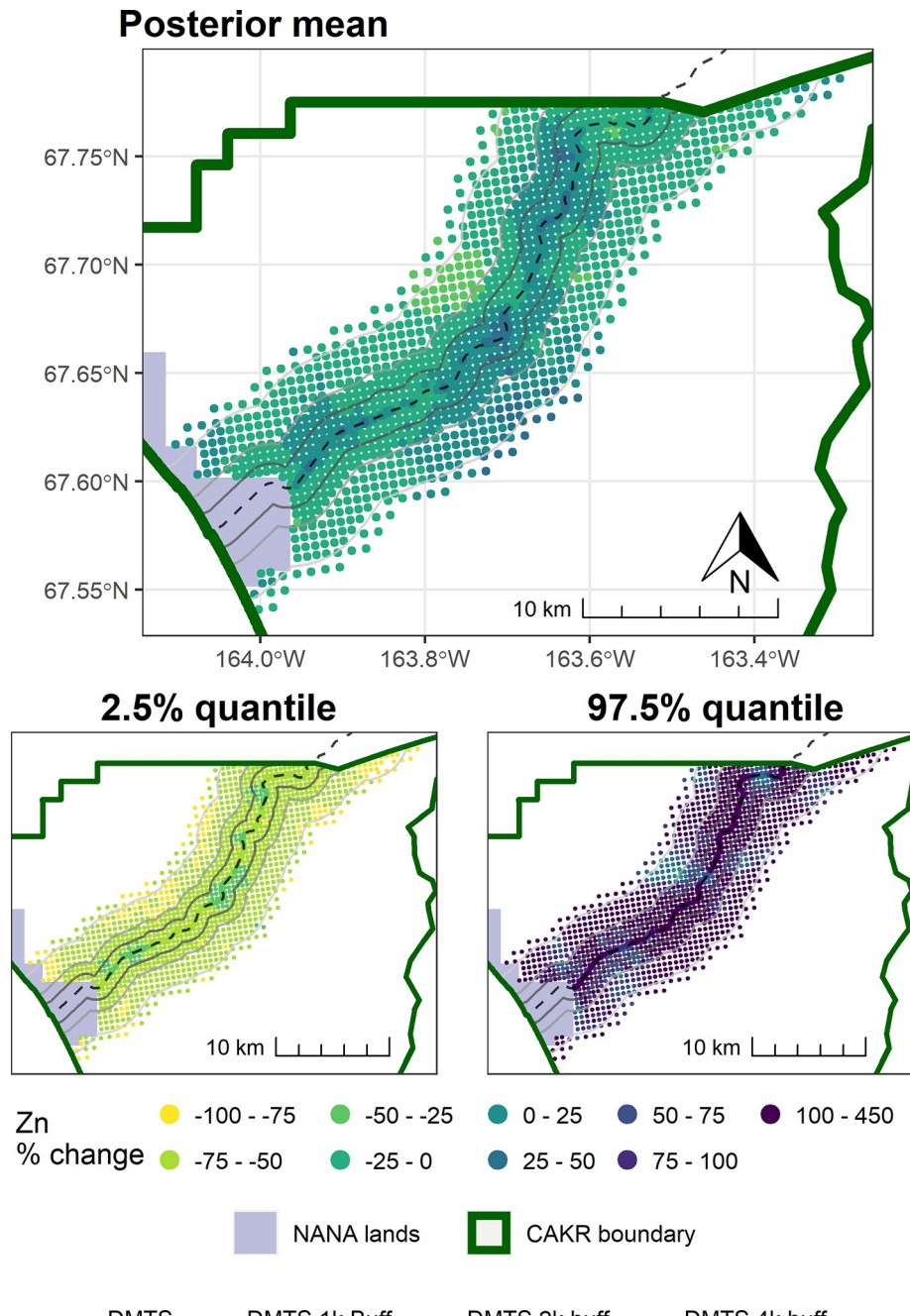

**Fig 5. Percent change in Zn concentrations in moss tissue in CAKR along the DMTS haul road, 2006–2017.**
Percent change between 2006 and 2017 Zn concentrations in moss tissue along the DMTS. The 2.5th and 97.5th
percentiles (lower and upper bound of the 95% credible interval) of the modeled concentrations are shown in smaller
maps. DMTS haul road and distance-based buffers at 1 km, 2 km, and 4 km are shown along with NANA lands and
the CAKR boundary.

Other biological effects including reduction in lichen cover, reduction in bryophyte cover,
reduction in lichen heights, reduction in bryophyte frond width have been reported from the
study area [18, 27, 34], but thresholds for these effects have not yet been determined. After

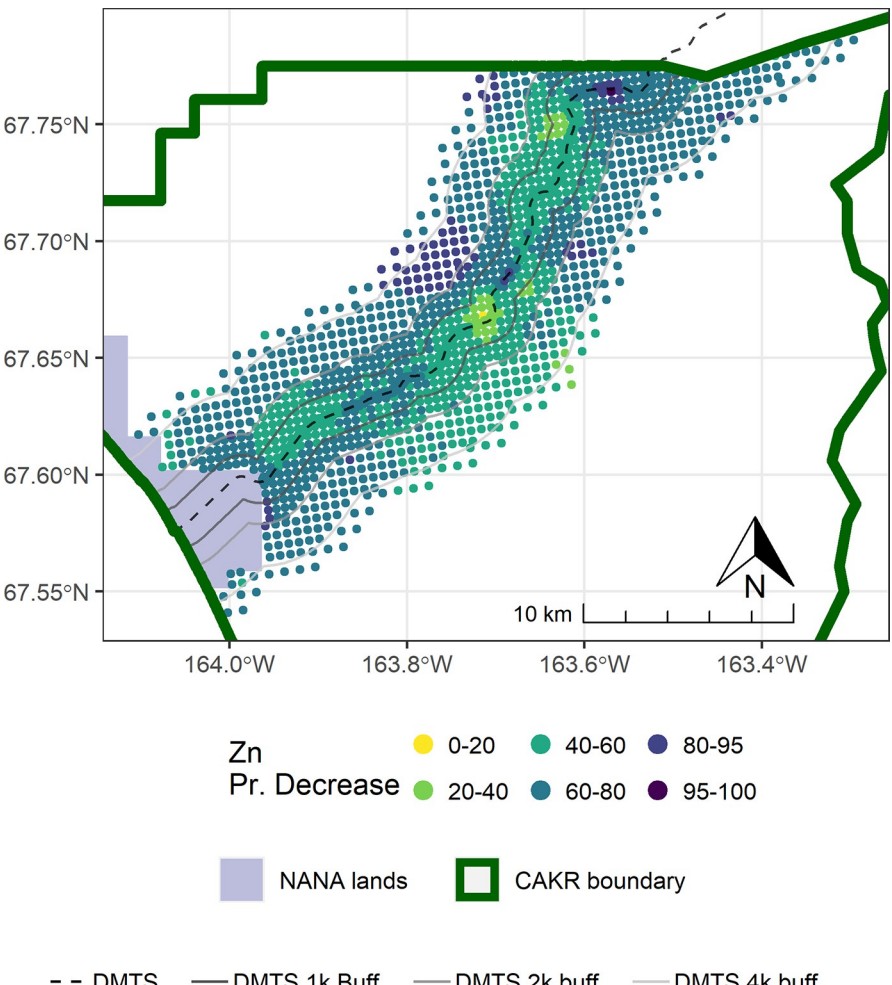

**Fig 6. Probability of Zn decrease in moss tissue in CAKR along the DMTS haul road, 2006–2017.** Posterior probability of decrease between 2006 and 2017 of Zn concentrations in moss tissue at each prediction point. DMTS haul road and distance-based buffers at 1 km, 2 km, and 4 km are shown along with NANA lands and the CAKR boundary.

thresholds for Zn by itself or in concert with Pb and Cd are defined for various biological effects, the results of this study may be used to model these effects spatially at the landscape level in CAKR. Moreover, because moss biomonitoring is relatively cheap and efficient relative to detailed biological field work, thresholds analysis may permit effective future monitoring simply via elemental analysis and spatial modeling of the elemental concentrations of moss samples.

There have been hundreds of studies in the past 5 decades using both lichens and bryophytes as bioindicators of air pollutants including N, S, semi-volatile organic compounds, and heavy metals [7, 35, 36]. Like this study, the majority have focused on quantifying levels of pollutants in lichen or moss tissue, and to a lesser extent, mapping deposition patterns [10]. The National Park Service monitoring project underlying the current research [19] is unique in examining both spatial patterns of heavy metals deposition on a sub-regional scale and linkages between pollution levels and mortality of lichens and bryophytes. In the context of the current literature, research on contaminants from mining-related fugitive

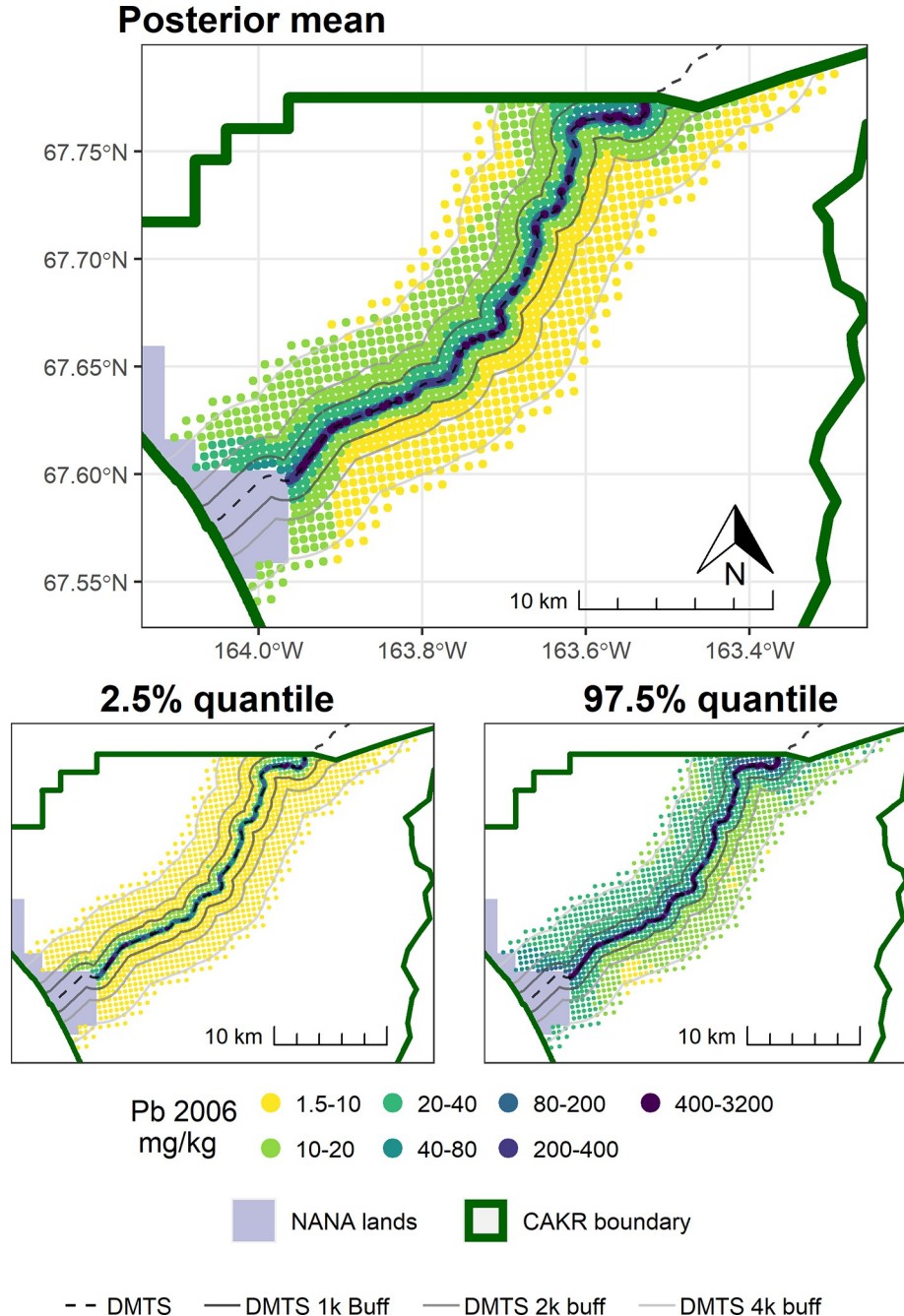

**Fig 7. Modeled 2006 Pb concentrations in moss tissue in CAKR along the DMTS haul road.** Smaller maps show the 2.5th and 97.5th percentiles (lower and upper bound of the 95% credible interval) of the modeled concentrations. DMTS haul road and distance-based buffers at 1 km, 2 km, and 4 km are shown along with NANA lands and the CAKR boundary.

dusts and their biological effects are uncommon compared to biomonitoring of urban and industrial point sources, smelters, regional air pollution, and soils polluted by mining wastes. The research comprising nearly 20 years of monitoring and modeling of fugitive dusts from the DMTS haul road in CAKR [4–6, 18, 27, 37, 38] represents an intensive case study on this distinct issue.

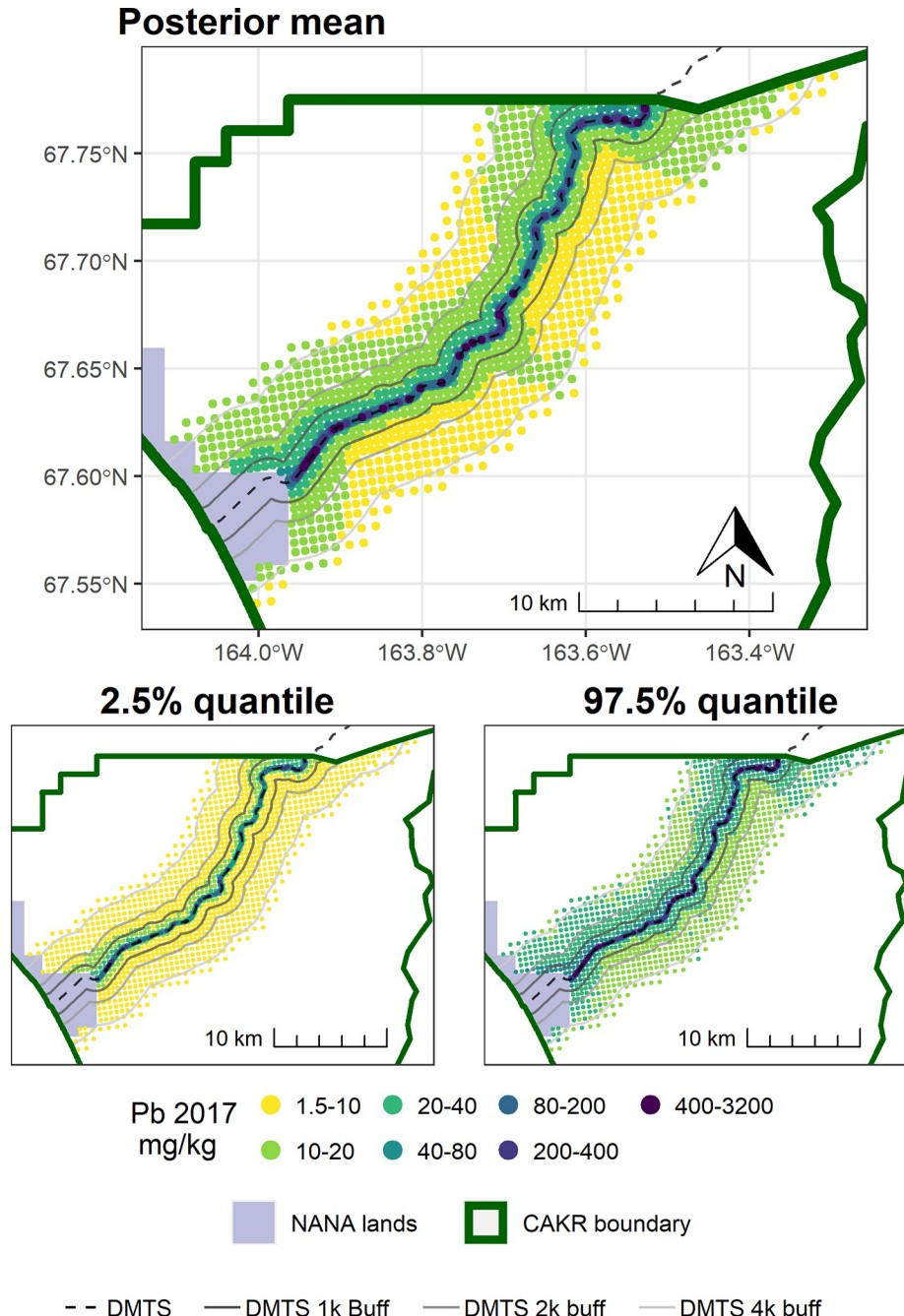

**Fig 8. Modeled 2017 Pb concentrations in moss tissue in CAKR along the DMTS haul road.** Smaller maps show the 2.5th and 97.5th percentiles (lower and upper bound of the 95% credible interval) of the modeled concentrations. DMTS haul road and distance-based buffers at 1 km, 2 km, and 4 km are shown along with NANA lands and the CAKR boundary.

**Mitigation.** Monitoring between 2001–2006 showed that the Red Dog Mine's mitigation during this period helped to reduce levels of Zn, Pb, and Cd in fugitive dusts significantly [4–6]. Major mitigations in during this period included replacing tarp-covered concentrate haul trucks with a fleet of trucks using hydraulically-covered lids, implementing a truck rinse station for use in summer months, containing escapement from the concentrate storage buildings

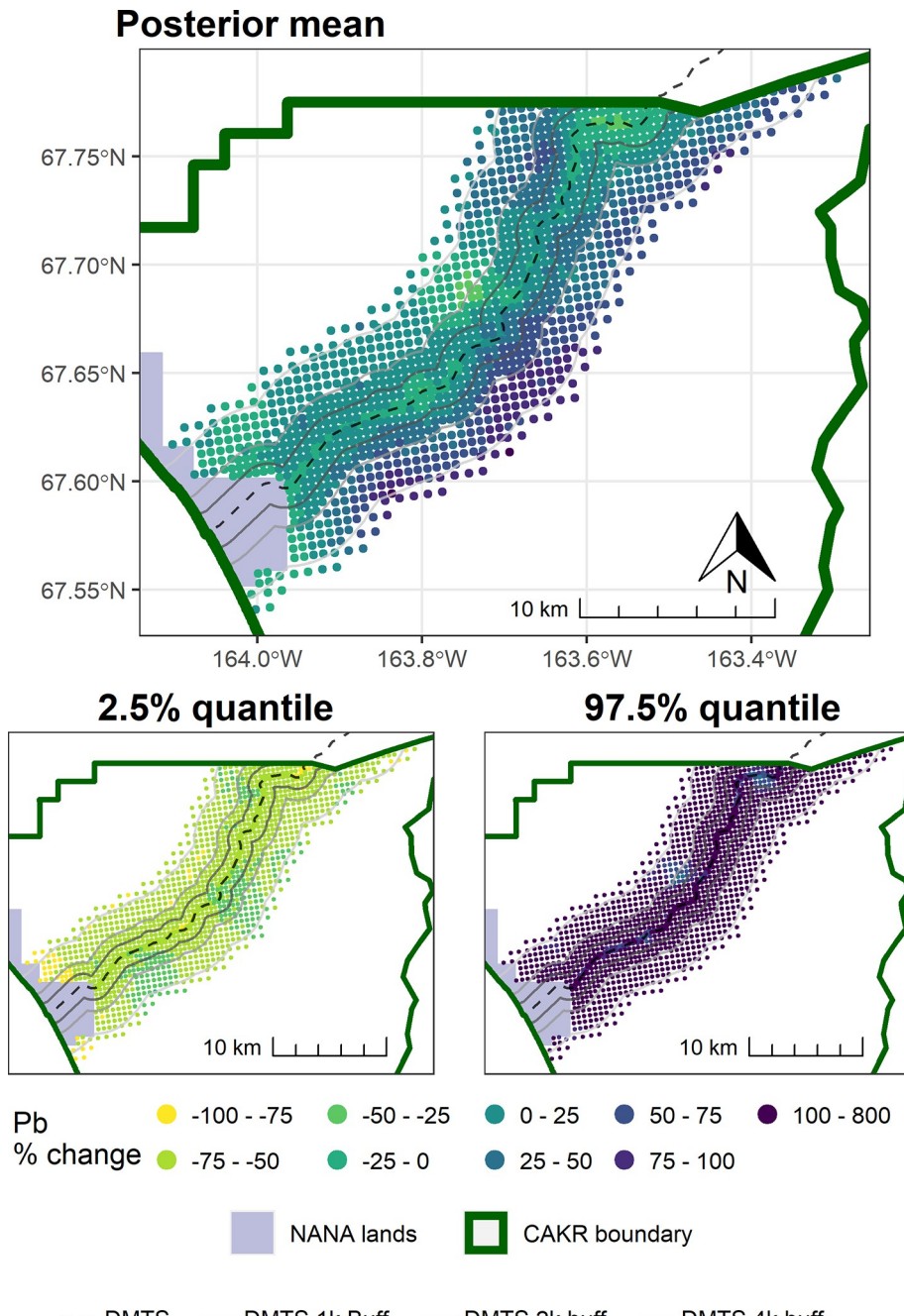

**Fig 9. Percent change in Pb concentrations in moss tissue in CAKR along the DMTS haul road, 2006–2017.**
Percent change between 2006 and 2017 Pb concentrations in moss tissue along the DMTS. The 2.5th and 97.5th
percentiles (lower and upper bound of the 95% credible interval) of the modeled concentrations are shown in smaller
maps. DMTS haul road and distance-based buffers at 1 km, 2 km, and 4 km are shown along with NANA lands and
the CAKR boundary.

via sealed loading and unloading facilities, enclosing the conveyor system that moves concentrates from the concentrate storage buildings to vessels at the Port Site, applying dust palliatives to the roadbed, and segregating road traffic to reduce concentrate tracking by vehicles [27]. The period between 2006–2017 showed improvements in some areas and slight worsening or

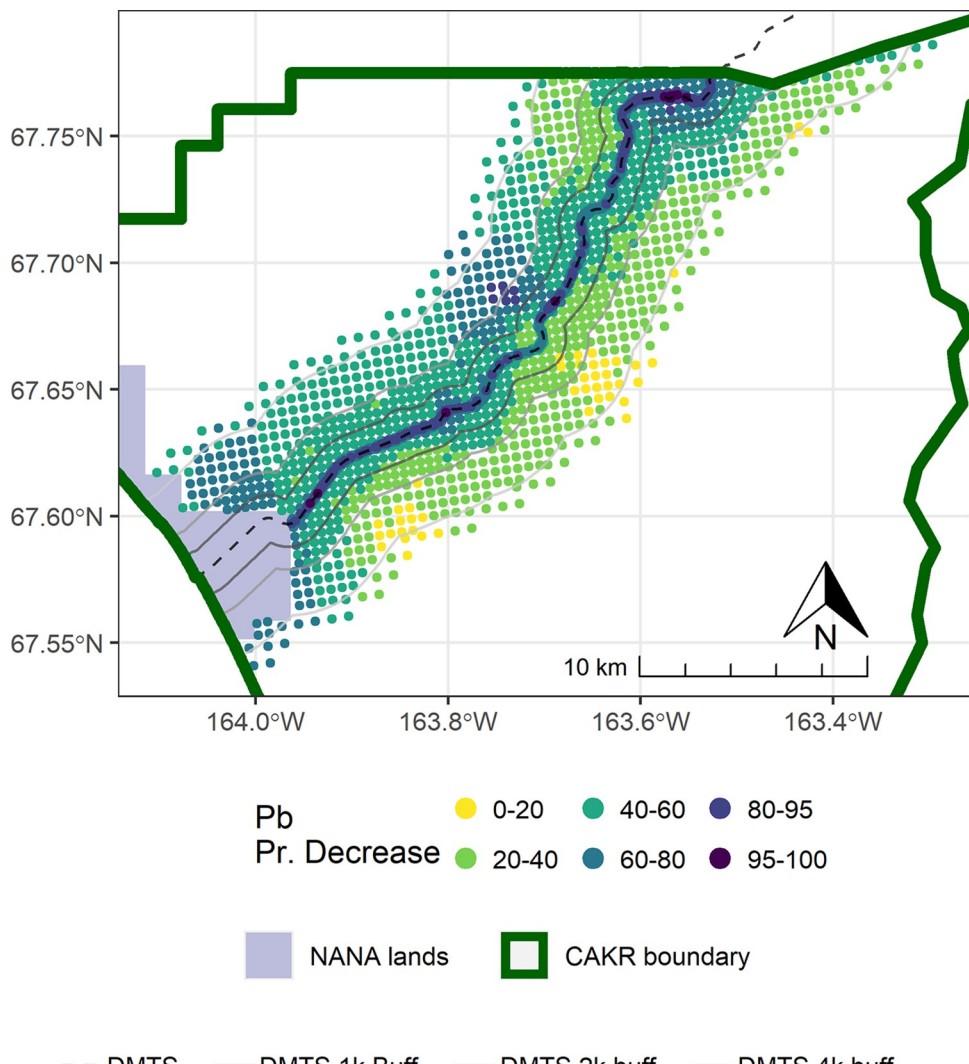

**Fig 10. Probability of Pb decrease in moss tissue in CAKR along the DMTS haul road, 2006–2017.** Posterior probability of decrease between 2006 and 2017 of Pb concentrations in moss tissue at each prediction point. DMTS haul road and distance-based buffers at 1 km, 2 km, and 4 km are shown along with NANA lands and the CAKR boundary.

no change in others, but the changes were generally much smaller than between 2001–2006. For instance, in Stratum 1, Zn dropped by 666 mg/kg between 2001–2006, a decrease of 54% [4]; between 2006–2017 there was no significant change in Zn concentrations (Fig 2 and Table 3). Similarly in Strata 2–4, there were decreases between 15–88 mg/kg of Zn between 2001–2006, while from between 2006–2017 decreases were limited to between 12–15 mg/kg of Zn. The success of mitigations from the 2001–2006 period has not resulted in a continuation of large decreases between 2006–2017. If greater reductions are sought to minimize the lichen and bryophyte declines reported in 2006 and 2017 [18, 27, 34], additional mitigations appear to be necessary. Study of surface peat soils along the DMTS haul road showed elevated levels of Zn, Pb, and Cd between 10–100 m from the road, with means of 484–785 mg/kg of Zn in organic soils [25]. Additional mitigation would slow the accumulation of heavy metals in these upper organic soil horizons.

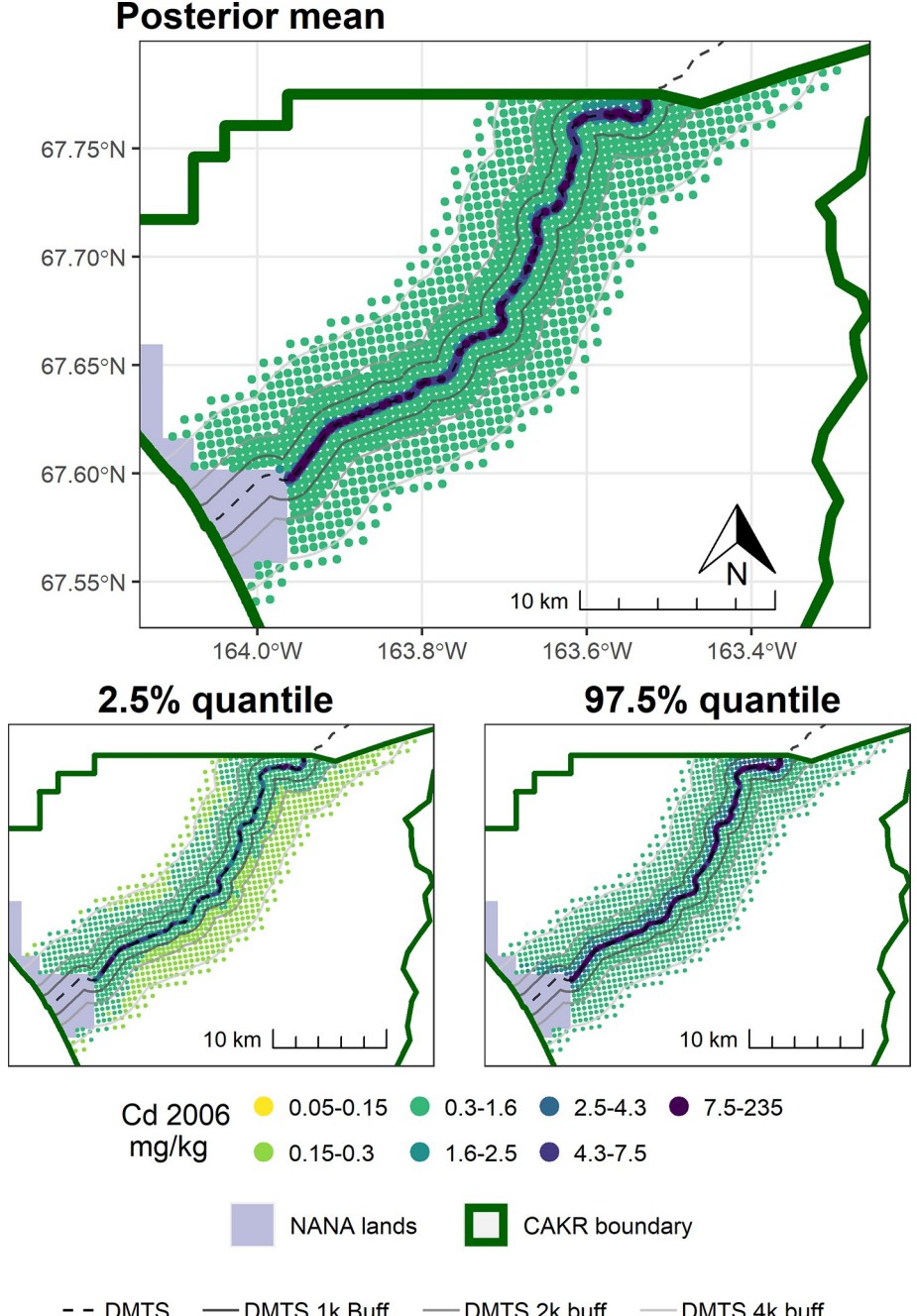

**Fig 11. Modeled 2006 Cd concentrations in moss tissue in CAKR along the DMTS haul road.** Smaller maps show the 2.5th and 97.5th percentiles (lower and upper bound of the 95% credible interval) of the modeled concentrations. DMTS haul road and distance-based buffers at 1 km, 2 km, and 4 km are shown along with NANA lands and the CAKR boundary.

## Conclusions

For moss tissue samples collected closest to the road (Stratum 1), only Pb decreased between 2006–2017 unlike the decreases in Zn, Pb, and Cd seen between 2001–2006. By contrast, between 2006–2017, both Zn and Cd decreased in at distances between 100–5,000 m (Strata

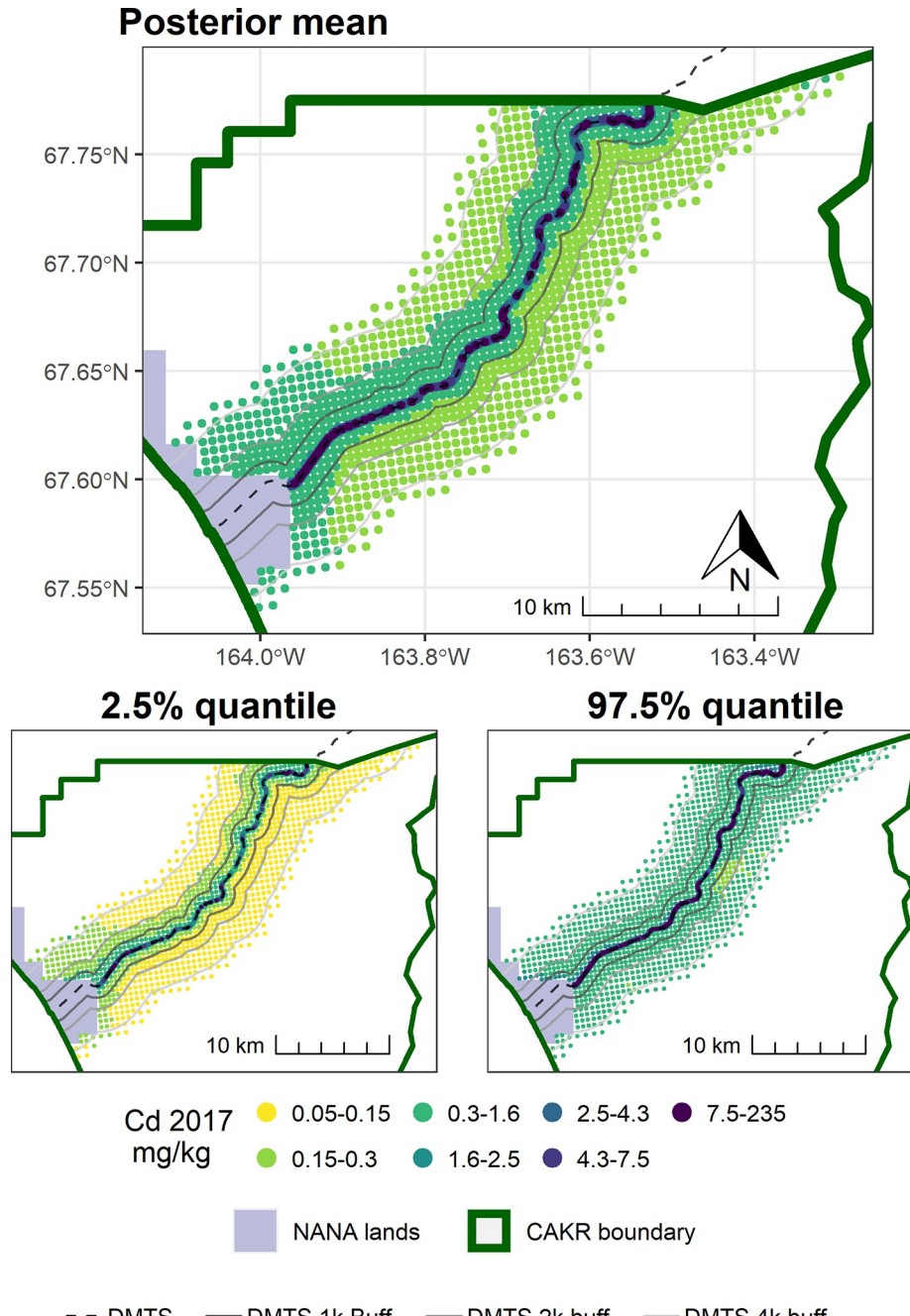

**Fig 12. Modeled 2006 Cd concentrations in moss tissue in CAKR along the DMTS haul road.** Smaller maps show the 2.5[th] and 97.5[th] percentiles (lower and upper bound of the 95% credible interval) of the modeled concentrations. DMTS haul road and distance-based buffers at 1 km, 2 km, and 4 km are shown along with NANA lands and the CAKR boundary.

2,3, and 4), with high probabilities of decrease and percent decreases of 11–20% and 46–52% respectively. Decreases in elemental concentrations in all strata were much more modest between 2006–2017 than between 2001–2006. Decreases in Zn during the most recent reporting period put Zn concentrations for a sizeable portion of the more distant reaches of the

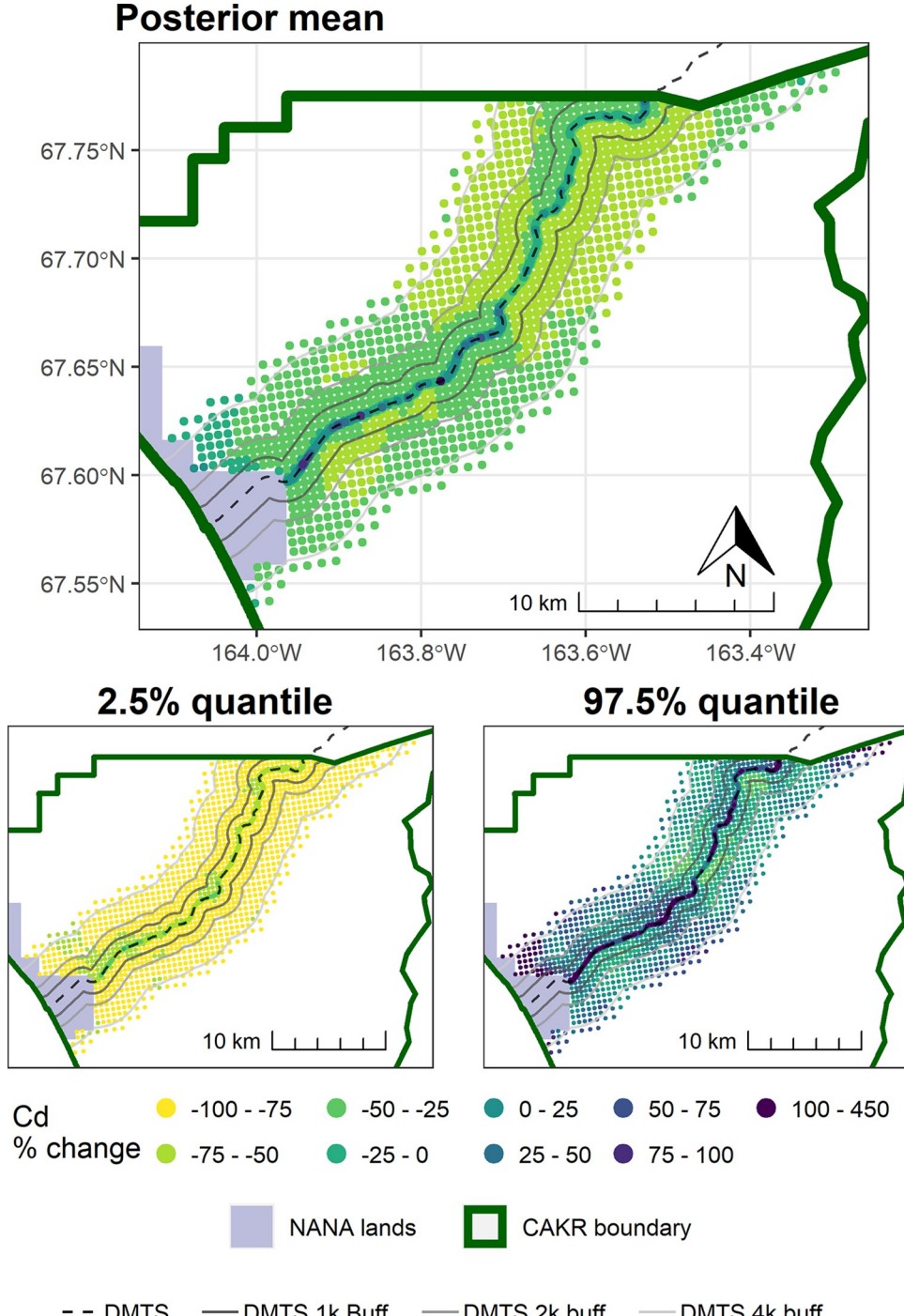

**Fig 13. Percent change in Cd concentrations in moss tissue in CAKR along the DMTS haul road, 2006–2017.**
Percent change between 2006 and 2017 Cd concentrations in moss tissue along the DMTS. The 2.5th and 97.5th percentiles (lower and upper bound of the 95% credible interval) of the modeled concentrations are shown in smaller maps. DMTS haul road and distance-based buffers at 1 km, 2 km, and 4 km are shown along with NANA lands and the CAKR boundary.

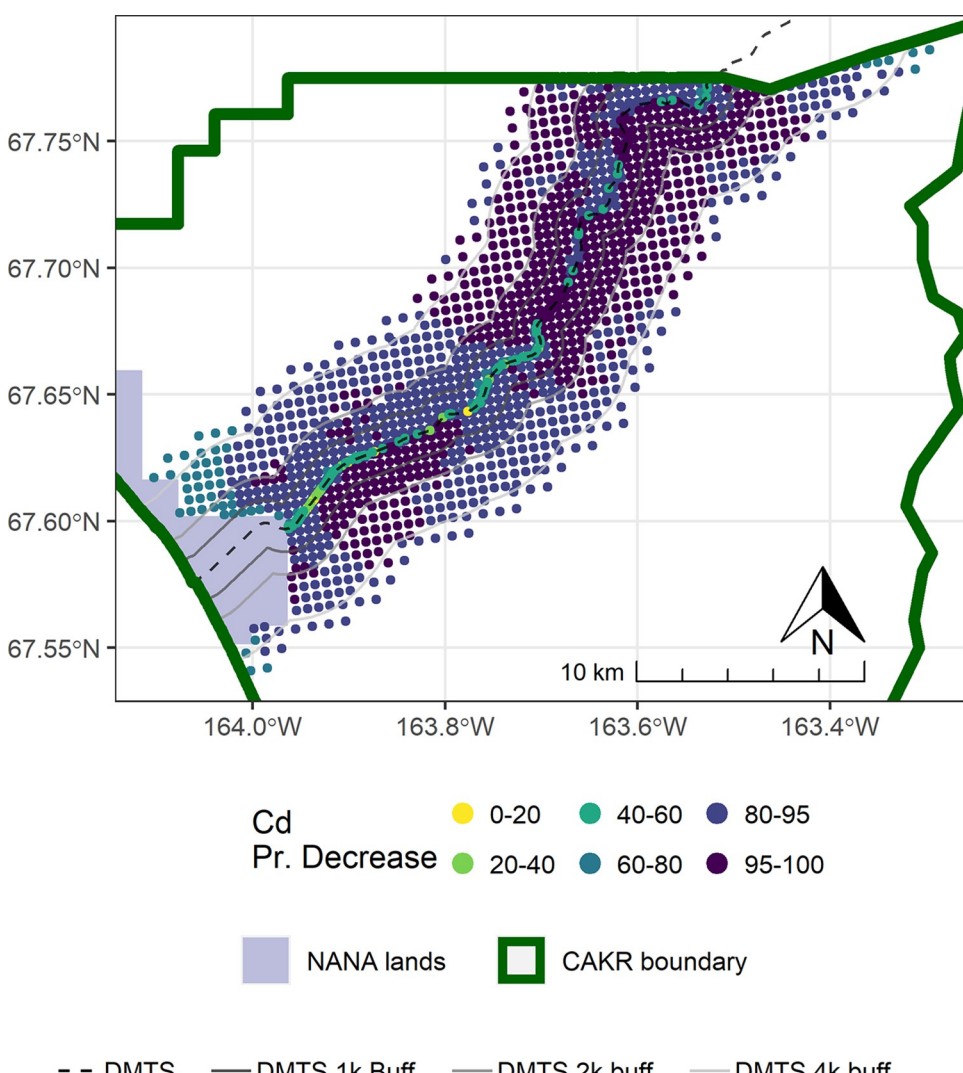

**Fig 14. Probability of Cd decrease in moss tissue in CAKR along the DMTS haul road, 2006–2017.** Posterior probability of decrease between 2006 and 2017 of Cd concentrations in moss tissue at each prediction point. DMTS haul road and distance-based buffers at 1 km, 2 km, and 4 km are shown along with NANA lands and the CAKR boundary.

study area close to (or below) the preliminary impact threshold (approximately 60 to 70 mg/kg of Zn) for lichen species richness. More work on the contaminant thresholds associated with different biological parameters (e.g., lichen species richness, lichen cover, nonvascular plant height and vigor) is needed. Lichen recovery is conceivable in areas that have dropped below the preliminary Zn threshold. However, the levels of Zn at which recovery would begin and the timeline for recovery will require ongoing field study of lichen communities and contaminant levels. The next National Park Service remeasurement of both vegetation and contaminants in moss tissue is scheduled for 2027.

## Acknowledgments

This study was conducted under the auspices of the National Park Service Arctic Network (Inventory and Monitoring Program). We thank Linda Hasselbach for the use of the 2001

elemental concentration data. Bruce McCune's lab at Oregon State University supported Elisa DiMeglio's master's thesis work (which included data collection for this project) and provided conceptual input. We thank Kali Melby and Elizabeth Rutila for their dedicated field work in 2017.

## Author Contributions

**Conceptualization:** Peter N. Neitlich, Wilson Wright, Alyssa E. Shiel, Mevin B. Hooten.

**Data curation:** Peter N. Neitlich, Wilson Wright, Elisa Di Meglio, Alyssa E. Shiel.

**Formal analysis:** Wilson Wright, Elisa Di Meglio, Alyssa E. Shiel, Celia J. Hampton-Miller.

**Funding acquisition:** Peter N. Neitlich.

**Investigation:** Peter N. Neitlich, Wilson Wright, Elisa Di Meglio, Alyssa E. Shiel.

**Methodology:** Peter N. Neitlich, Wilson Wright, Elisa Di Meglio, Alyssa E. Shiel, Mevin B. Hooten.

**Project administration:** Peter N. Neitlich, Elisa Di Meglio, Alyssa E. Shiel, Mevin B. Hooten.

**Resources:** Peter N. Neitlich.

**Supervision:** Peter N. Neitlich, Elisa Di Meglio, Alyssa E. Shiel, Mevin B. Hooten.

**Validation:** Peter N. Neitlich, Wilson Wright, Elisa Di Meglio, Mevin B. Hooten.

**Visualization:** Peter N. Neitlich, Wilson Wright, Mevin B. Hooten.

**Writing – original draft:** Peter N. Neitlich, Wilson Wright, Elisa Di Meglio, Celia J. Hampton-Miller.

**Writing – review & editing:** Peter N. Neitlich, Wilson Wright, Elisa Di Meglio, Alyssa E. Shiel, Celia J. Hampton-Miller, Mevin B. Hooten.

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
