## [Decision Letter · Decision Letter 0]

14 Mar 2023

PONE-D-23-00195Mixed trends in heavy metal-enriched fugitive dust on National Park Service lands along the Red Dog Mine haul road, Alaska, 2006–2017.PLOS ONE

Dear Dr. Neitlich,

Thank you for submitting your manuscript to PLOS ONE. After careful consideration, we feel that it has merit but does not fully meet PLOS ONE’s publication criteria as it currently stands. Therefore, we invite you to submit a revised version of the manuscript that addresses the points raised during the review process.

Dear Author, your paper is extremely interesting but academically not robust enough to be accepted for publication as it is. For this reasons I asked four reviewers rather than only two, to help the journal making a decision. They made very good comments and after reading the paper myself, I believe that your methodology is the weakest part of it. Given the nature and impact of some of the remarks (especially those made by Reviewer 1), I am not entirely sure you can fix the paper, but I believe that the topic you are addressing is quite important and deserves another chance. Before re-submitting, please, thoroughly address all the remarks raised at your best.

We look forward to receiving your revised manuscript.

Kind regards,

Maurizio Fiaschetti

Academic Editor

PLOS ONE

Journal Requirements:

“This study was funded by the National Park Service’s Arctic Network (Inventory and Monitoring Program).”

“This research was funded by the National Park Service, Arctic Network (Inventory and Monitoring Program).”

“This research was funded by the National Park Service, Arctic Network (Inventory and Monitoring Program).”

6. We note that Figures 1, 3, 4, 5, 6, 7, 8, 9, 10, 11, 12, 13 and 14 in your submission contain map images which may be copyrighted. All PLOS content is published under the Creative Commons Attribution License (CC BY 4.0), which means that the manuscript, images, and Supporting Information files will be freely available online, and any third party is permitted to access, download, copy, distribute, and use these materials in any way, even commercially, with proper attribution. For these reasons, we cannot publish previously copyrighted maps or satellite images created using proprietary data, such as Google software (Google Maps, Street View, and Earth). For more information, see our copyright guidelines: http://journals.plos.org/plosone/s/licenses-and-copyright.

 a. You may seek permission from the original copyright holder of Figures 1, 3, 4, 5, 6, 7, 8, 9, 10, 11, 12, 13 and 14 to publish the content specifically under the CC BY 4.0 license. 

Reviewers' comments:

Reviewer's Responses to Questions

**Comments to the Author**

1. Is the manuscript technically sound, and do the data support the conclusions?

Reviewer #1: No

Reviewer #2: Partly

Reviewer #3: Yes

Reviewer #4: Yes

2. Has the statistical analysis been performed appropriately and rigorously? 

Reviewer #1: N/A

Reviewer #2: Yes

Reviewer #3: Yes

Reviewer #4: Yes

3. Have the authors made all data underlying the findings in their manuscript fully available?

Reviewer #1: Yes

Reviewer #2: No

Reviewer #3: Yes

Reviewer #4: Yes

4. Is the manuscript presented in an intelligible fashion and written in standard English?

Reviewer #1: Yes

Reviewer #2: Yes

Reviewer #3: Yes

Reviewer #4: Yes

5. Review Comments to the Author

Reviewer #1: Dear Authors:

I have revised the manuscript PONE-D-23-00195 and I found that this work lack of a well-documented and properly supported hypothesis. Data are just too much descriptive and the weight of the entire manuscript relays merely in statistics. No other kind of discussion regarding the extremely important biological and environmental factors that for sure are impacting in the level of metals detected in the studied area are even proposed. For example, biogenic sources of metals, the yield of moss biomass during time, the impact of climate change in moss physiology and at the studied area (e.g., rainfall rate, rainfall intensity, sun irradiation, etc.), the life cycle of these specimens, the duplicating time, the viability and physiological affection due to heavy metals accumulation, type of sediments, quality of soil, etc. etc. etc.… none of these aspects (or at least some of them) are considered in this manuscript. Therefore, without any other aim more than the statistics itself, from my point of view, conclusions are not meaningful at all. I am truly sorry, but in its present form I cannot recommend this article to be publish in PLOS ONE.

Reviewer #2: The paper prepared by Peter N. Neitlich and co-authors looks like a report and not a scientific paper. Usually, scientific papers are not written in the first person!

All parts of the manuscript need to be significantly improved.

1. Introduction: The novelty of the study needs to be added. Study is of very local interest, how the results can be used by international community?

2. Methods: how many samples were collected for analysis, which techniques were used to determine elements concentration of the elements?

3. Results and Discussion: in the manuscript there presented results, but there is no discussion

Reviewer #3: Dear Author,

I found your manuscript to be well-written and concise. However, I suggest that you provide more details on the sampling and laboratory analyses methods used, to add to the clarity of the manuscript.

Thank you.

Reviewer #4: The authors presented the results of a decadal spatial analysis of contaminant deposition based on the resampling of locations Cape Krusenstern National Monument (CAKR), Alaska, adjacent to the Red Dog Mine haul road. There are several comments to be solved by the authors, after that I can recommend its publication in PLOS ONE.

- The purpose of this study is not very clear stated in the abstract. Please add the importance of your findings also.

- Please describe more in Field and Lab Methods how many samples did you collect. How did you keep these until the analyses? What is the unit of measurement for your assays?

- I suggest to comment your results in comparison with other available studies in this domain of research, and related to the human activities organized in that region. The paper has no discussions in this moment.

- Regarding your last phrase in the Conclusions, did you measure the lichen recovery? Can you support your statement?

6. PLOS authors have the option to publish the peer review history of their article (what does this mean?). If published, this will include your full peer review and any attached files.

Reviewer #1: No

Reviewer #2: No

Reviewer #3: No

Reviewer #4: No

---

## [Author Response · Author response to Decision Letter 0]

8 Jun 2023

Please see Response to Reviewers for our detailed responses.

---

## [Decision Letter · Decision Letter 1]

20 Jul 2023

PONE-D-23-00195R1Mixed trends in heavy metal-enriched fugitive dust on National Park Service lands along the Red Dog Mine haul road, Alaska, 2006–2017.PLOS ONE

Dear Dr. Neitlich,

Thank you for submitting your manuscript to PLOS ONE. After careful consideration, we feel that it has merit but does not fully meet PLOS ONE’s publication criteria as it currently stands. Therefore, we invite you to submit a revised version of the manuscript that addresses the points raised during the review process.

The manuscript has greatly improved, but still not quite suitable for publication. I encourage the authors to address the issues raised by Reviewer 5 and in particular, I encourage them to provide a more thorough discussion of the methodoloy used and the results obtained. All the abbreviations need to be explained too.

We look forward to receiving your revised manuscript.

Kind regards,

Maurizio Fiaschetti

Academic Editor

PLOS ONE

Journal Requirements:

Reviewers' comments:

Reviewer's Responses to Questions

**Comments to the Author**

1. If the authors have adequately addressed your comments raised in a previous round of review and you feel that this manuscript is now acceptable for publication, you may indicate that here to bypass the “Comments to the Author” section, enter your conflict of interest statement in the “Confidential to Editor” section, and submit your "Accept" recommendation.

Reviewer #4: All comments have been addressed

Reviewer #5: (No Response)

2. Is the manuscript technically sound, and do the data support the conclusions?

Reviewer #4: Yes

Reviewer #5: Partly

3. Has the statistical analysis been performed appropriately and rigorously? 

Reviewer #4: Yes

Reviewer #5: N/A

4. Have the authors made all data underlying the findings in their manuscript fully available?

Reviewer #4: Yes

Reviewer #5: Yes

5. Is the manuscript presented in an intelligible fashion and written in standard English?

Reviewer #4: Yes

Reviewer #5: Yes

6. Review Comments to the Author

Reviewer #4: (No Response)

Reviewer #5: Your manuscript seems interesting because it raises an important environmental issue. The layout of the manuscript is standard for scholarly articles. The introduction and purpose were written correctly. Unfortunately, the next chapters need improvement. The methodology is not clear and does not reflect the Results and Discussion chapter. There is no explanation of the abbreviations under the tables. The discussion of the results is cursory and general. Literature should not be cited in the conclusion.

The language appears to be correct, but I don't feel qualified to judge about the English language and style.

good luck!

Sincerely yours

Reviewer

7. PLOS authors have the option to publish the peer review history of their article (what does this mean?). If published, this will include your full peer review and any attached files.

Reviewer #4: No

Reviewer #5: No

---

## [Author Response · Author response to Decision Letter 1]

2 Nov 2023

Response to Reviewers

From: Peter Neitlich, National Park Service

To: Dr. Maurizio Fiaschetti, Academic Editor, PLOS ONE

Re: Response to Reviewers

Date: 

Dear Dr. Fiaschetti,

Thank you for forwarding the excellent suggestions for revisions to this manuscript. These comments have helped to create a much better version of the final product. I have responded to each revision suggestion and have summarized resulting revisions in the comment matrix below. I am confident that we captured the essence of the requested revisions, but please do let me know if you still see any unresolved issues.

Best wishes,

Peter Neitlich

Reviewer Comment Revision

Academic Editor I encourage the authors to address the issues raised by Reviewer 5 and in particular, I encourage them to provide a more thorough discussion of the methodology used and the results obtained. 

Authors: We have greatly enhanced the methods section and have also pointed readers who want more information to open source publications that report methods of this long-term monitoring protocol in granular detail.

Academic Editor: All the abbreviations need to be explained too. 

Authors: Completed.

Reviewer 5: Your manuscript seems interesting because it raises an important environmental issue. The layout of the manuscript is standard for scholarly articles. The introduction and purpose were written correctly. Unfortunately, the next chapters need improvement. The methodology is not clear and does not reflect the Results and Discussion chapter.

Authors: We have expanded the methods section and have also pointed readers who want more information to open source publications in this journal that report methods of this long-term monitoring protocol in high detail. We have also explained our goals in the methods better such that they align with the Results and Discussion.

Reviewer 5: There is no explanation of the abbreviations under the tables. 

Authors: All abbreviations have now been defined in each table legend.

Reviewer 5: The discussion of the results is cursory and general. 

Authors: We have considerably enhanced the discussion of biological effects and have discussed analogous literature on recolonization of lichens following pollution reduction.

Reviewer 5: Literature should not be cited in the conclusion. 

Authors: We have moved this idea and citation to the Results and Discussion section.

---

## [Decision Letter · Decision Letter 2]

12 Jan 2024

Mixed trends in heavy metal-enriched fugitive dust on National Park Service lands along the Red Dog Mine haul road, Alaska, 2006–2017.

PONE-D-23-00195R2

Dear Dr. Neitlich,

We’re pleased to inform you that your manuscript has been judged scientifically suitable for publication and will be formally accepted for publication once it meets all outstanding technical requirements.

Kind regards,

Maurizio Fiaschetti

Academic Editor

PLOS ONE

Reviewers' comments:

Reviewer's Responses to Questions

**Comments to the Author**

1. If the authors have adequately addressed your comments raised in a previous round of review and you feel that this manuscript is now acceptable for publication, you may indicate that here to bypass the “Comments to the Author” section, enter your conflict of interest statement in the “Confidential to Editor” section, and submit your "Accept" recommendation.

Reviewer #5: All comments have been addressed

2. Is the manuscript technically sound, and do the data support the conclusions?

Reviewer #5: Yes

3. Has the statistical analysis been performed appropriately and rigorously? 

Reviewer #5: Yes

4. Have the authors made all data underlying the findings in their manuscript fully available?

Reviewer #5: Yes

5. Is the manuscript presented in an intelligible fashion and written in standard English?

Reviewer #5: Yes

6. Review Comments to the Author

Reviewer #5: (No Response)

7. PLOS authors have the option to publish the peer review history of their article (what does this mean?). If published, this will include your full peer review and any attached files.

Reviewer #5: No

---

## [Editor Report · Acceptance letter]

17 Feb 2024

PONE-D-23-00195R2 

PLOS ONE

Dear Dr. Neitlich, 

I'm pleased to inform you that your manuscript has been deemed suitable for publication in PLOS ONE. Congratulations! Your manuscript is now being handed over to our production team.

Kind regards, 

on behalf of

Dr. Maurizio Fiaschetti 

Academic Editor

PLOS ONE